

# 1 Photochemical Degradation of Isoprene-derived 4,1-Carbonyl
# 2 Nitrate

F. Xiong[1], C. H. Borca[1], L. V. Slipchenko[1] and P. B. Shepson[1,2]
[1] Department of Chemistry, Purdue University, West Lafayette, IN
[2] Department of Earth, Atmospheric and Planetary Sciences, Purdue University, West
Lafayette, IN
Correspondence to: P. B. Shepson (pshepson@purdue.edu)
**Abstract**
In isoprene-impacted environments, carbonyl nitrates are produced from $NO_3$-initiated isoprene
oxidation, which constitutes a potentially important $NO_x$ reservoir. To better understand the fate
of isoprene carbonyl nitrates, we synthesized a model compound, *trans*-4-nitrooxy-2-methyl-2-
buten-1-al (4,1-isoprene carbonyl nitrate) and investigated its photochemical degradation process.
The measured OH and $O_3$ oxidation rate constants for this carbonyl nitrate are $4.1(\pm0.7)\times10^{-11}$
$cm^3$ molecules$^{-1}$ s$^{-1}$ and $4.4(\pm0.3)\times10^{-18}$ $cm^3$ molecules$^{-1}$ s$^{-1}$. The UV absorption spectrum of the
carbonyl nitrate was determined, and the result is consistent with TDDFT calculations. Based on
its UV absorption cross section and photolysis frequency in a reaction chamber, we estimate that
the ambient photolysis frequency for this compound is $3.1\times10^{-4}$ s$^{-1}$ for a solar zenith angle
(SZA) of 45°. The fast photolysis rate and high reactivity toward OH lead to a lifetime of less
than one hour for the carbonyl nitrate, with photolysis being a dominant daytime sink. The
nitrate products derived from the OH oxidation and the photolysis of the isoprene carbonyl
nitrate were identified with an iodide-based chemical ionization mass spectrometer. For the OH
oxidation reaction, we quantified the yields of two nitrate products, MVK nitrate and ethanal
nitrate, which together contributed to 37(±5)% of the first-generation products.
**1 Introduction**
Over the past century, tropospheric ozone concentrations have increased from around 20 ppb to
~40 ppb, with urban-impacted concentrations often rising to 60-100 ppb (Parrish et al., 2014;
Vingarzan, 2004), posing harmful effects on human health and crop yields (Lefohn and Foley,





1993; Lippmann, 1989). Tropospheric ozone is catalytically produced in the chemical reactions
of nitrogen oxides ($NO_x \equiv NO + NO_2$) and volatile organic compounds (VOCs) (Haagen-Smit,
1952). $NO_2$ photolysis forms ozone (Blacet, 1952), and the ozone production rate is enhanced
when the $NO$-$NO_2$-$O_3$ cycle is coupled with the oxidation of VOCs (Chameides et al., 1988;
Chameides and Walker, 1973; Chameides et al., 1992). When $NO_x$ is incorporated into organic
molecules and forms organic nitrates ($RONO_2$), however, ozone formation is suppressed
(Roberts, 1990). Organic nitrates are a temporary $NO_x$ reservoir. Degradation of organic nitrates
can release $NO_2$ back into the atmosphere (Aschmann et al., 2011), and thus facilitate ozone
production. Organic nitrates in the gas phase can also adsorb onto atmospheric aerosols,
followed by condensed-phase hydrolysis (Rindelaub et al., 2015). This process removes the
reactive nitrogen from the atmosphere permanently, as the nitrooxy group is turned into the non-
volatile $NO_3^-$ ion (Darer et al., 2011; Hu et al., 2011). The relative importance of these parallel
nitrate sinks affects the availability of $NO_x$ and the ozone production rate in the troposphere.
Therefore, detailed understanding of the loss mechanisms of organic nitrates is crucial to
understanding the dynamics of ground-level ozone formation.
Modeling studies suggest that isoprene-derived organic nitrates have substantial influence on the
$NO_x$ cycle and tropospheric $O_3$ production (Horowitz et al., 2007; Mao et al., 2013; Paulot et al.,
2012; Wu et al., 2007). During the daytime, isoprene is lost rapidly to OH oxidation, forming
organic nitrates through the $RO_2 + NO$ reaction with a yield of 7-14% (Lockwood et al., 2010;
Patchen et al., 2007; Paulot et al., 2009; Sprengnether et al., 2002; Tuazon and Atkinson, 1990;
Xiong et al., 2015). At night, reaction with $NO_3$ is a significant removal pathway for isoprene
(Brown et al., 2009; Starn et al., 1998), and organic nitrates constitute 65-70% of the oxidation
products (Perring et al., 2009; Rollins et al., 2009). While $NO_3$-initiated isoprene oxidation
contributes to a small fraction of isoprene loss, this reaction pathway could generate
approximately half of the isoprene-derived organic nitrates on a regional scale, due to its large
nitrate yield (Horowitz et al., 2007; Xie et al., 2013).
Fig.1 shows the formation pathways of organic nitrate products from $NO_3$-initiated oxidation of
isoprene, including hydroperoxy nitrate, carbonyl nitrate and hydroxy nitrate. Reactions for only
one of the nitrooxy peroxy radicals are shown for brevity. The hydroxy nitrates can be also
formed in the OH-initiated isoprene oxidation reactions, and their production and degradation



have been studied extensively in both laboratory and field studies (Chen et al., 1998; Giacopelli
et al., 2005; Grossenbacher et al., 2004; Jacobs et al., 2014; Lee et al., 2014b; Lockwood et al.,
2010; Patchen et al., 2007; Paulot et al., 2009; Sprengnether et al., 2002; Tuazon and Atkinson,
1990; Xiong et al., 2015). For the hydroperoxy nitrates, Schwantes et al. (2015) investigated
their production from the $RO_2 + HO_2$ reaction and identified the nitrooxy hydroxyepoxide
product from the OH oxidation of the isoprene hydroperoxy nitrate. For the isoprene carbonyl
nitrates, their formation has been quantified in an experimental study (Kwan et al., 2012), but
their sinks and fate can only be inferred from analog molecules, such as nitrooxy ketones, due to
lack of direct studies on these specific compounds. Suarez-Bertoa et al. (2012) conducted
kinetics experiments on three synthesized saturated nitrooxy ketones, and their results indicate
that photolysis is the dominant sink for these nitrate compounds. By comparing the published
UV absorption spectra of α-nitrooxy ketones with the UV spectra of the mono-functional nitrates
and ketones, Müller et al. (2014) suggested that the nitrooxy ketones have enhanced absorption
cross sections, which can facilitate the dissociation of the O−$NO_2$ bond. Like the nitrooxy
ketones discussed by Suarez-Bertoa et al. (2012) and Müller et al. (2014), the carbonyl nitrate
derived from $NO_3$ + isoprene oxidation has a chromophore, −C=C−C=O, at the β position of the
nitrate group, which may enhance the UV absorption cross section of the molecule and facilitate
its photolytic dissociation. However, since the five-carbon isoprene carbonyl nitrate (Fig. 1) is
unsaturated, it is expected to be lost rapidly to OH oxidation. To date, the relative importance of
the individual photochemical sinks for the unsaturated carbonyl nitrates is still unclear. To
answer this question, we synthesized a model compound for the five-carbon isoprene carbonyl
nitrates, and investigated its photochemical reactivities.

## 81   **2   Synthesis and characterization**

A model compound, 4,1-isoprene carbonyl nitrate (*trans*-2-methyl-4-nitrooxy-2-buten-1-al ) was
synthesized following the reaction scheme in Fig. 2. The nitrate was prepared by reacting $AgNO_3$
with the corresponding bromide (*trans*-4-bromo-2-methyl-2-buten-1-al) (Ferris et al., 1953),
which was synthesized following Gray (1981). The $^1H$ and $^{13}C$ NMR spectra of the synthesized
product are shown in Fig. S1 and Fig. S2. Its IR absorption spectrum is shown in Fig. S3.



Shown in Fig. 3 are the UV absorption cross sections for the carbonyl nitrate, methacrolein
(MACR) and isopropyl nitrate, obtained in acetonitrile solvent. The absorption cross section of
the carbonyl nitrate is enhanced relative to that of MACR, but the two spectra have similar
features from 320 nm to 400 nm with peak absorption at 325 nm. This is probably because they
both contain the O=C–C=C chromophore. Below 320 nm the absorption of the carbonyl nitrate is
enhanced significantly in comparison with that of isopropyl nitrate. This observation is
consistent with reports from Müller et al. (2014) that molecules containing α,β-nitrooxy ketone
functionalities have enhanced UV absorption.
**3   Methods**
**3.1   Setup for the kinetics chamber experiments**
Three sets of reaction chamber experiments were conducted to determine the photolysis
frequency, OH oxidation rate constant and the $O_3$ oxidation rate constant for the carbonyl nitrate.
The experiments were performed in the 5500 L Purdue photochemical reaction chamber (Chen et
al., 1998). A chemical ionization mass spectrometer (CIMS) with $I^-$ as the reagent ion was used
to quantify the carbonyl nitrate and its nitrate degradation products (Xiong et al., 2015). The
chamber air was sampled into the CIMS through a 5.2 m long FEP tubing (0.8 cm ID, heated to
constant 50 °C). The photolysis frequenecy was obtained by measuring the loss of the carbonyl
nitrate inside the reaction chamber in the presence of UV radiation and propene as a radical
scavenger. When the UV lamps were turned off, the wall loss rate constant for the carbonyl
nitrate was dervied by observing its slow decay, with propene as an ozone and $NO_3$ scavenger.
The OH reaction rate constant and $O_3$ reaction rate constant were obtained using the relative rate
method  (Atkinson and Aschmann, 1985). Propene was used as the reference compound, and its
changing concentrations were measured using a GC-FID equipped with a 0.32 mm Rtx-Q-Bond
column. For the OH oxidation experiments, OH was generated through the photolysis of
isopropyl nitrite, which was synthesized follwoing Noyes (1933). NO was added to the chamber
to suppress the formation of $O_3$. In addition, two OH oxidation experiments were performed
without propene in order to quantify the oxidation products. For the OH-initiated oxidation
experiments, NO and $NO_2$ were measured using the Total REactive Nitrogen Instrument
(TRENI) (Lockwood et al., 2010). The ozonolysis experiments were performed in the dark, and



cyclohexane was added to the chamber as an OH scavenger. The initial conditions for the
experiments are listed in Table S1.

## 3.2  Computational methods

The theoretical UV absorption spectra of the carbonyl nitrate, MACR, and *n*-butyl nitrate in the
gas phase were calculated and analyzed, in four stages, using time-dependent density functional
theory (TDDFT; Hohenberg and Kohn, 1964; Kohn and Sham, 1965; Runge and Gross, 1984).
All calculations were carried out using the computational chemistry package Q-Chem 4.3 (Shao
et al., 2015). First, the structure of each molecule was optimized employing the long-range
corrected hybrid density functional ωB97X-D (Chai and Head-Gordon, 2008) with the 6-31+G*
basis set (Frisch et al., 1984). A high accuracy grid was employed. Second, frequencies
calculations were executed on the optimized structures to verify their accuracy. These were run
using the same setup described above. Third, after assuring the structures represented adequate
minima, the first ten singlet excited states of each molecule were computed with TDDFT, using
the same functional and basis set. Finally, a visual analysis of the molecular orbitals (MOs) was
carried out with the visualization software IQmol 2.7 (Gilbert, 2012).

## 4  Results

### 4.1  Absorption spectra and density functional calculations

Fig. 4 shows the TDDFT UV absorption spectra of the carbonyl nitrate, MACR, and *n*-butyl
nitrate. There are three groups of transitions in the simulated spectra.
Both MACR and the carbonyl nitrate show a relatively weak transition in the region around 330
nm, which corresponds to the first electronic transition, from the highest occupied molecular
orbital (HOMO) to the lowest unoccupied molecular orbital (LUMO), in both molecules. Fig. 5a
provides comparative information between the first electronic transition of the carbonyl nitrate
and the homologous excitation of MACR. As shown in Fig. 5a the character of the molecular
orbitals involved in this transition is similar in both cases, indicating that the aldehyde group is
involved in the first electronic excitation of the carbonyl nitrate.



Fig. 5b shows the information corresponding to the second electronic transition of the carbonyl
nitrate and the homologous excitation of *n*-butyl nitrate. Both transitions are found in the region
around 255 nm, and they are 3 orders of magnitude darker than those at 330 nm. Inspection of
the character of the MOs involved in these processes reveals a correspondence between the
second electronic excitation of the carbonyl nitrate, HOMO−2 → LUMO+1, and the HOMO →
LUMO transition of *n*-butyl nitrate. As with the previous case, that observation confirms that the
nitrate group is involved in the second electronic excitation of the carbonyl nitrate, but at
wavelengths shorter than present at the Earth's surface. Fig. 5b also shows that in this case, the
local character of the MOs involved in the transition is even more pronounced, with bulky lobes
placed mainly over the nitrate group.
Even though the second electronic transition of carbonyl nitrate is not displayed in the
experimental spectra of Fig. 3, because its range covers from 280 nm to 410 nm, it is reasonable
to assume that it is caused by the local excitation of the nitrate group, based on the computational
results. Thus, it can be suggested that the experimental UV absorption spectra of isopropyl
nitrate is comparable to that of *n*-butyl nitrate simulated computationally. Thus it is possible that
the feature in the region around 280 nm of the carbonyl nitrate experimental spectrum of Fig. 3
could be caused by a broadening of the transition located around 255 nm.
The brightest transition in the modeled spectra of the carbonyl nitrate, 3 orders of magnitude
brighter than the ones at 330 nm, is located around 210 nm. There are two transitions in this
region and each one has a homologous excitation: the HOMO−1 → LUMO in carbonyl nitrate is
similar to HOMO−1 → LUMO in MACR, and the HOMO−5 → LUMO+1 in carbonyl nitrate is
related to the (mainly) HOMO−1 → LUMO transition of *n*-butyl nitrate. These transitions are
beyond the range of the experimental spectra on Fig. 3 and beyond the atmospherically relevant
absorption wavelengths.
**4.2 Photochemical sinks of 4,1-carbonyl nitrate**
Fig. 6 shows the first-order wall loss and photolysis loss of the carbonyl nitrate inside the
reaction chamber. The wall loss rate constant was $1.3(\pm 0.1)\times 10^{-5}$ s$^{-1}$, and the photolysis rate
constant was $3.0(\pm 0.1)\times 10^{-5}$ s$^{-1}$, after subtracting the wall loss rate constant from the first-order
decay rate constant measured for the photolysis experiments. It is worth mentioning that our





reactant carbonyl nitrate has a *trans* configuration, and it may photo-isomerize into the *cis*
configuration, which would be detected at the same m/z by the CIMS. The cis-carbonyl nitrate
can either photo-dissociate or isomerize to re-form the *trans* isomer. Our previous work suggests
that the CIMS is 4 times more sensitive to the *cis* configuration than the *trans* configuration
(Xiong et al., 2015). If a significant amount of the *cis* isomer was present, the CIMS signal
should resemble a double exponential curve, because the *cis* isomer was being produced and
consumed simultaneously. In an extreme scenario with rapid *trans* → *cis* isomerization, the
CIMS signal should increase under radiation, due to the higher sensitivity of the *cis* isomer. For
our carbonyl nitrate photolysis experiments, a single exponential decay in the CIMS signal was
observed, indicating insignificant contribution from the *cis* isomer. Hence, our measured
photolysis frequency should well characterize the loss rate of the carbonyl nitrate inside the
reaction chamber.
Since the UV radiation inside the reaction chamber is different from the UV radiation in the
ambient environment (Fig. 7), $Cl_2$ was used as a reference compound to extrapolate the nitrate
photolysis rate from chamber radiation to solar radiation. The photolysis decay of $Cl_2$ in the
reaction chamber was measured with the CIMS (Neuman et al., 2010). Cyclohexane was added
to the chamber to scavenge the Cl atoms so that $Cl_2$ was not re-formed from Cl + Cl
recombination. The first-order photolysis rate constant for $Cl_2$ was $2.50(\pm0.04)\times10^{-4}$ $s^{-1}$ (Fig.
S4).
The photolysis frequency (J) is the integrated product of quantum yield (Φ), absorption cross
section (σ, $cm^2$) and actinic flux (F, $cm^{-2}$ $s^{-1}$) across all wavelengths (Eq. 1). Therefore, the
photolysis frequencies for the carbonyl nitrate and $Cl_2$ in the reaction chamber can be compared
as in Eq. 2.
$J = \int \Phi_\lambda \sigma_\lambda F_\lambda d\lambda$ (Eq. 1)
$\dfrac{J_{Cl_2}^{chamber}}{J_{nitrate}^{chamber}} = \dfrac{\sum \varphi_{Cl_2} \sigma_{Cl_2} F_{chamber}}{\sum \varphi_{nitrate} \sigma_{nitrate} F_{chamber}}$ (Eq. 2)
$J_{Cl_2}^{chamber}$ and $J_{nitrate}^{chamber}$ are the photolysis frequencies of $Cl_2$ and the carbonyl nitrate inside the
chamber. $\sigma_{Cl_2}$ and $\sigma_{nitrate}$ are the cross sections for $Cl_2$ and the carbonyl nitrate at each





wavelength. $\sigma_{nitrate}$ was determined by this work (Fig. 3). $\sigma_{Cl_2}$ has been measured previously
and the IUPAC recommended values were used (Atkinson et al., 2007). $F_{chamber}$ is the
wavelength-dependent flux of photons inside the chamber. The radiation spectrum (Fig. 7) of the
chamber UV lamps (UVA 340) was obtained from the manufacturer (Q-lab), but the actual
absolute radiation intensity in the chamber is expected to differ from the manufacturer's
radiation spectrum by a scaling factor, because of the inverse-square dependence on distance,
and our specific multi-lamp geometry. When $Cl_2$ was used as a reference compound for the
nitrate photolysis rate, the scaling factors in Eq. 2 will cancel.
The Cl-Cl bond dissociation energy is 243 kJ/mol (Luo, 2007a), equivalent to a photon at 492
nm. Since $Cl_2$ has only one bond, it has unity quantum yield below 492 nm and zero quantum
yield above 492 nm. The emission spectrum of the UV lamps for the reaction chamber is
centered from 300 nm to 400 nm (Fig. 7). Hence, $\varphi_{Cl_2}$ =1 in Eq. 2, at all wavelengths. For the
carbonyl nitrate, however, its quantum yield is affected by the bond dissociation energy,
intramolecular vibrational energy redistribution and relaxation of the excited molecule from
collisions, so an average effective quantum yield ($\varphi_{nitrate}^{eff}$) is assumed, and Eq. 2 becomes Eq. 3.
Since the photolysis rates, absorption cross sections and chamber radiation spectrum were known,
we calculated that $\varphi_{nitrate}^{eff}$ was 0.48.
$$\frac{J_{Cl_2}^{chamber}}{J_{nitrate}^{chamber}} = \frac{\sum \sigma_{Cl_2} F_{chamber}}{\varphi_{nitrate}^{eff} \sum \sigma_{nitrate} F_{chamber}} \qquad \text{(Eq. 3)}$$
The effective quantum yield of 0.48 indicates that when the carbonyl nitrates absorbs a photon
inside the reaction chamber, the probability (averaged across the absorption wavelengths) for it
to dissociate is 48%. However, the probability for nitrate photolysis is not equal at all
wavelengths, the low energy photons (long wavelength) being less likely to induce photo-
dissociation. Hence, we introduced a threshold wavelength $\lambda_0$, for which the carbonyl nitrate has
unity quantum yield below $\lambda_0$ and zero quantum yield above $\lambda_0$. Although this approach accounts
for the energy difference of photons with different wavelengths, it is still a very rough
estimation. Using the threshold wavelength, the effective quantum yield can be expressed by Eq.
4 and Eq. 5, where $\varphi(\lambda)$ is the quantum yield of the carbonyl nitrate, and $F(\lambda)$ is the chamber



photon flux (Fig. 7), as a function of the wavelength $\lambda$. Solving for the unknown $\lambda_0$ in Eq. 5, we
calculated that $\lambda_0$ was 347 nm.
$$\varphi(\lambda) = \begin{cases} 1 \ (\lambda \leq \lambda_0) \\ 0 \ (\lambda > \lambda_0) \end{cases}$$         (Eq. 4)
$$\frac{\sum_\lambda F(\lambda) \cdot \varphi(\lambda)}{\sum_\lambda F(\lambda)} = 0.48$$         (Eq. 5)
The solar radiation spectrum was calculated with the TUV model (Madronich and Flocke, 1998).
By assuming that the carbonyl nitrate has zero quantum yield above 347 nm and unity quantum
yield below 347 nm, its photolysis frequency is $2.6 \times 10^{-4}$ s$^{-1}$ for a solar zenith angle (SZA) of
45°, and $3.7 \times 10^{-4}$ s$^{-1}$ for SZA of 0°. It is worth mentioning that the condensed-phase and gas-
phase absorption spectra should be different, because the solvent molecules affect the
polarization and dipole moment of the solute (Bayliss and McRae, 1954; Braun et al., 1991;
Linder and Abdulnur, 1971). Although we were unable to measure the gas-phase cross section of
the carbonyl nitrate, we could assess the uncertainty caused by using the condensed-phase
spectrum in our calculation, by comparing the gas-phase and condensed-phase spectra of MACR
and isopropyl nitrate (Fig. S5a). On average, the gas-phase absorption cross sections of MACR
and isopropyl nitrate are 1.7 times those in the solution phase (Fig. S5b). For the carbonyl nitrate,
if the gas-phase cross section is assumed to be 1.7 times that of the solution-phase cross section,
the calculated effective quantum yield becomes 0.28, leading to a threshold wavelength ($\lambda_0$) of
336 nm. Using this set of cross section and quantum yields, we calculated that the nitrate
photolysis frequency was $3.1 \times 10^{-4}$ s$^{-1}$ for SZA of 45°, and $4.6 \times 10^{-4}$ s$^{-1}$ for SZA of 0°, which are
19% and 24% larger than results obtained using the condensed-phase cross section. The
calculated ambient photolysis frequency is not affected as significantly by the change in the
absorption cross section, because it is constrained by the measured photolysis frequency in the
reaction chamber. When a larger cross section is applied, a smaller quantum yield is derived, and
the calculated ambient photolysis frequency, being the integrated product of the cross section,
quantum yield and radiation, will not increase as much as the cross section. In addition to the
cross section, our treatment of the wavelength-dependent quantum yield can also introduce
uncertainty to the calculated results. If a constant effective quantum yield is used in the
calculation, the ambient photolysis frequency is $2.0 \times 10^{-4}$ s$^{-1}$ for SZA of 45°, and $2.8 \times 10^{-4}$ s$^{-1}$ for





SZA of 0°, which are 23% and 24% lower than assuming a threshold wavelength. Therefore, our
calculated ambient photolysis frequency, based on condensed-phase absorption cross section and
a threshold energy for unity quantum yield, has an uncertainty of 25%. Since we believe that the
cross sections are indeed larger in the gas phase, our best estimate is $3.1 \times 10^{-4}$ s$^{-1}$ for SZA=45°.
Fig. 8 shows the results for the relative rate experiments for the OH-initiated and O$_3$-initiated
oxidation of the carbonyl nitrate, with propene as the reference compound. The loss of the
carbonyl nitrate to wall uptake and photolysis is corrected when comparing the oxidative loss of
the nitrate to that of propene, using the same method as Hallquist et al. (1997). The OH and O$_3$
oxidation rate constants for propene are $3.0(\pm 0.5) \times 10^{-11}$ cm$^3$ molecules$^{-1}$ s$^{-1}$ (Klein et al., 1984;
Zellner and Lorenz, 1984) and $1.00(\pm 0.06) \times 10^{-17}$ cm$^3$ molecules$^{-1}$ s$^{-1}$ (Herron and Huie, 1974;
Treacy et al., 1992). These are the IUPAC preferred rate constants for T=298K
(http://iupac.pole-ether.fr/). Hence, the OH and O$_3$ oxidation rate constants for the isoprene
carbonyl nitrate are, based on the results from the relative rate experiments, $4.1(\pm 0.7) \times 10^{-11}$ cm$^3$
molecules$^{-1}$ s$^{-1}$ and $4.4(\pm 0.3) \times 10^{-18}$ cm$^3$ molecules$^{-1}$ s$^{-1}$ respectively, at 295 K.
The OH oxidation rate constant for the carbonyl nitrate can be estimated through the structure-
activity-relationship (SAR) approach proposed by Kwok and Atkinson (1995). The rate constant
for OH addition to the double bond can be calculated as k(-CH=CH), which is $8.69 \times 10^{-11}$ cm$^3$
molecules$^{-1}$ s$^{-1}$, multiplied by the two correction factors C(-CHO) and C(-CH$_2$ONO$_2$), which are
0.34 and 0.47 respectively. The resulting OH addition rate constant is $1.39 \times 10^{-11}$ cm$^3$
molecules$^{-1}$ s$^{-1}$. The rate constant for H abstraction from the –CHO group is $1.61 \times 10^{-11}$ cm$^3$
molecules$^{-1}$ s$^{-1}$, after multiplying a correction factor of 1 for having a double bond at its α
position. The rate constant for H abstraction from the methylene group is $3.7 \times 10^{-14}$ cm$^3$
molecules$^{-1}$ s$^{-1}$, calculated by multiplying the base rate constant for methylene groups, which is
$9.34 \times 10^{-13}$ cm$^3$ molecules$^{-1}$ s$^{-1}$, by the correction factors of the nitrate group and the double
bond, which are 0.04 and 1, respectively. OH addition to the nitrate group has a rate constant of
$4.4 \times 10^{-13}$ cm$^3$ molecule$^{-1}$ s$^{-1}$, after taking account of the enhancement factor of 1.23 for the
methylene group. H abstraction from the methyl group has a rate constant of $1.36 \times 10^{-13}$ cm$^3$
molecules$^{-1}$ s$^{-1}$. By summing up the rate constants for all these reaction pathways, the SAR-
derived OH oxidation rate constant for the 4,1-carbonyl nitrate rate constant is $3.1 \times 10^{-11}$ cm$^3$
molecules$^{-1}$ s$^{-1}$, approximately 30% lower than the experimental measurement. The dominant





reaction channels are OH addition to the double bond and H abstraction from the aldehyde
group. Contributions from the other reaction pathways are small (<3%).
The relative importance of the three photochemical sinks, photolysis, OH oxidation and $O_3$
oxidation, depends on the solar radiation and the concentrations of OH and $O_3$. To better
illustrate their relative contributions, observations of OH and $O_3$ from previous field campaigns
were used to calculate the loss rates of the carbonyl nitrate. The local solar radiation was
calculated with the TUV model (Madronich and Flocke, 1998), which was then used to derive
the photolysis frequency. The calculated results (Fig. 9) suggest that photolysis is a significant
degradation pathway for the carbonyl nitrate, which can dominate over OH oxidation toward
mid-day. When the solar radiation intensity is small (such as 6:00 AM for the 1999 SOS
campaign), OH oxidation is likely the dominant sink. Due to the fast photolysis and high
reactivity toward OH, the photochemical lifetime of the carbonyl nitrate can be as short as less
than one hour.

### 296    4.3    Degradation products of the 4,1-carbonyl nitrate

### 297    4.3.1    OH oxidation

The products from the OH-initiated oxidation of the 4,1-carbonyl nitrate were observed by the
CIMS. The change in the CIMS signals before and after the reaction are illustrated in Fig. 10,
along with assignment of some of the molecular structures based on the molecular weight and
likely chemistry. The OH-initiated oxidation reaction can proceed through two channels: H
abstraction from the aldehyde group and OH addition to the double bond.
For the H abstraction pathway, a peroxyacyl nitrate (PAN) product was observed at m/z 349 (Fig.
10), which can be formed as shown in Fig. 11. The first-order dissociation rate constant for the
PAN compound was determined at room temperature (295 K) using the following method. A
100 L Teflon bag containing the air mixture of approximately 1 ppm isopropyl nitrite and 30 ppb
4,1-carbonyl nitrate was irradiated, and the PAN compound was formed from OH and $NO_2$
(produced through the photolysis of isopropyl nitrite) reaction with the 4,1-carbonyl nitrate.
After 5 min reaction time, the bag was removed from the UV radiation, and NO was injected into
the bag to around 4 ppm in concentration. The bag was then sampled simultaneously by the
CIMS, which monitored the decrease in the signal of the PAN compound, and by the TRENI,





which monitored the concentrations of NO and NO$_2$. The PAN dissociation reaction is a
reversible process, where the dissociation products, peroxyacyl (PA) radical and NO$_2$, can re-
combine to form PAN. With the addition of the large amount of NO, PA radicals are
predominantly consumed by the irreversible PA + NO reaction, leading to the decay of the PAN
compound. The apparent PAN dissociation rate constant can be described by Eq. 6 (Shepson et
al., 1992), where k is the first-order loss rate constant measured by the CIMS (Fig. S6), $k_{PAN}$ is
the real PAN dissociation rate constant, [NO] and [NO$_2$] are the concentrations for NO and NO$_2$,
and $k_{NO}$ and $k_{NO2}$ are the rate constants for PA + NO and PA + NO$_2$ reactions. Since the rate
constants $k_{NO}$ and $k_{NO2}$ for the carbonyl nitrate-derived PA radical are unknown, the IUPAC
recommended rate constants for the peroxyacetyl radicals (CH$_3$C(O)O$_2$) are used, with $k_{NO}$ =
$2.0 \times 10^{-11}$ cm$^3$ molecule$^{-1}$ s$^{-1}$ and $k_{NO2}$ = $8.9 \times 10^{-12}$ cm$^3$ molecule$^{-1}$ s$^{-1}$. The PAN dissociation
rate constant, after correcting for the competing PA + NO and PA + NO$_2$ reactions using Eq. 6, is
$5.7(\pm 0.8) \times 10^{-4}$ s$^{-1}$, based on three experimental trials. In addition to dissociation, the PAN
compound in the 100 L bag could also undergo wall loss. This loss rate was estimated by
multiplying the wall loss rate of the carbonyl nitrate in the 5500 L chamber by a factor of 16,
which is the square diffusion distance of the chamber relative to that of the 100 L bag, assuming
the PAN compound and the isoprene carbonyl nitrate have similar diffusion and adsorption
coefficients. Considering the uncertainty in wall loss rate, the PAN dissociation rate constant is
$5.7(+0.8/-2.8) \times 10^{-4}$ s$^{-1}$. Previous studies of the dissociation rate constants for peroxyacyl nitrates
have reported results ranging from $1.6 \times 10^{-4}$ s$^{-1}$ to $6.0 \times 10^{-4}$ s$^{-1}$ at 298 K (Bridier et al., 1991;
Grosjean et al., 1994; Kabir et al., 2014; Roberts and Bertman, 1992). Our result is consistent
with previous work.
$$k = k_{PAN}\left(1 - \frac{1}{1 + \frac{k_{NO}[NO]}{k_{NO_2}[NO_2]}}\right)$$     (Eq. 6)
Since our OH oxidation experiments were conducted in the presence of high NO concentration, a
significant fraction of the PA radicals from the H abstraction reaction channel were expected to
react with NO to form alkoxy radicals. Based on the product observed at m/z 321, a reaction
scheme (Fig. 11) is proposed, where the alkoxy radical dissociates into CO$_2$ and an alkyl radical,
which is further oxidized to form a C4 dinitrate (m/z 321, Fig. 10), along with ethanal nitrate
(m/z 232, Fig. 10).





For the OH addition pathway, OH can add to the C2 and the C3 position of the 4,1-isoprene
carbonyl nitrate, but the less substituted C3 position should be preferential (Peeters et al., 2007).
For the C2 addition, the expected nitrate products are C5 dinitrate and ethanal nitrate (Fig. 12a),
as observed at m/z 351 and m/z 232 (Fig. 10). $NO_2$ could potentially be released with the
concurrent formation of a C4 di-aldehyde (Fig. 12a). The CIMS signal for this compound at m/z
229 did not increase (Fig. 10), but the CIMS sensitivity for this compound could be relatively
low. For the C3 addition, the expected nitrate products are C5 dinitrate, MVK nitrate and ethanal
nitrate (Fig. 12b), observed at m/z 351, m/z 276 and m/z 232 (Fig. 10). The C2 and C3 OH
addition pathway would lead to two C5 dinitrate isomers, but they were detected at the same
mass by the CIMS.
Using a GC-ECD/CIMS method similar to the one described by Xiong et al. (2015), the CIMS
sensitivities of the nitrate products were determined relative to the CIMS sensitivity of the 4,1-
carbonyl nitrate. The setup was modified to operate the GC separation under pressure lower than
1 atm (Fig. S7), which helped to lower the elution temperature. A Teflon bag filled with the 4,1-
carbonyl nitrate, isopropyl nitrite, and NO was irradiated to generate the OH oxidation products.
The mixture of the 4,1-carbonyl nitrate and its products were then cryo-focused and separated on
the GC column, and eluent species were detected by the ECD and the CIMS simultaneously. We
were able to quantify the MVK nitrate and the ethanal nitrate using this method, assuming
identical ECD sensitivities for nitrates. The other products shown in Fig. 10, however, were not
detected with simultaneous good signal-to-noise ratio on the ECD and the CIMS. The
ECD/CIMS chromatograms are shown in Fig. 13. We determined that the reaction of the 4,1-
carbonyl nitrate and the reagent ion $I^-$ could form $NO_3^-$, but the same reaction did not occur for
the MVK nitrate and the ethanal nitrate (Fig. 13). Formation of $NO_3^-$ from $I^-$ reaction with
organic nitrates has not been reported previously. Since $I^-$ is a poor nucleophile, it is unclear if
this reaction proceeds by $S_N2$ substitution. Using the same $I^-$ ionization method, Wang et al.
(2014) observed $NO_3^-$ signal equivalent to a $NO_3 + N_2O_5$ concentration of 200-1000 ppt during a
field study in Hong Kong. Through interference tests, the authors attributed 30-50% of the
observed $NO_3^-$ signal to the interference from peroxyacetyl nitrate and $NO_2$. Since $I^-$ reaction
with the carbonyl nitrate can also generate $NO_3^-$, organic nitrates ($RONO_2$) could be a potential
source of interference for $NO_3 + N_2O_5$ measurement with the $I^-$ ionization method.





For the GC-ECD/CIMS calibration, 9 trials were conducted at three different pressures. The
results are summarized in Table S2. The relative CIMS sensitivities for the 4,1-carbonyl nitrate,
ethanal nitrate and MVK nitrate are 1:15(±3):34(±3) respectively. The absolute CIMS sensitivity
of the 4,1-carbonyl nitrate was determined with standard gas samples prepared following Xiong
et al. (2015), and the result was used to calculate the absolute sensitivities for the ethanal nitrate
and the MVK nitrate. The ethanal nitrate and the MVK nitrate both have the $-ONO_2$ group at the
β position of the acidic H, so their CIMS sensitivities are comparable. For the MVK nitrate, the
electron-withdrawing ketone group can further enhance its gas-phase acidity and its affinity to
bind with $I^-$. Hence, the CIMS sensitivity for the MVK nitrate is greater than for the ethanal
nitrate. For the 4,1-carbonyl nitrate, its low CIMS sensitivity can be caused by the *trans*-δ
configuration of the $-ONO_2$ group and the $-CHO$ group. Our previous studies on isoprene-
derived hydroxynitrates suggested that the CIMS sensitivity for the β isomer is 8 times greater
than for the *trans*-δ isomer (Xiong et al., 2015). Lee et al. (2014a) also reported the β isomer
sensitivity being over 16 times greater than the *trans*-δ isomer sensitivity, using iodide as the
reagent ion. Hence, our calibration results, with the sensitivity for the ethanal nitrate 15 times
greater than the sensitivity for the 4,1-carbonyl nitrate, is consistent with previous work.
With the CIMS sensitivities determined, the yield of the MVK nitrate and the ethanal nitrate
from the OH-initiated oxidation of 4,1-carbonyl nitrate was obtained by comparing the formation
of the products relative to the loss of the reactant (Fig. 14). The ethanal nitrate was corrected for
loss to OH oxidation and photolysis, using the method described by Tuazon et al. (1984). The
applied ethanal nitrate + OH rate constant was $3.4\times10^{-12}\,cm^3\,molecules^{-1}\,s^{-1}$, calculated using the
structure-reactivity relationship proposed by Kwok and Atkinson (1995). The photolysis
frequency of the isoprene carbonyl nitrate was applied to account for the photolytic loss of
ethanal nitrate inside the chamber, because the β-ketone group is known to enhance the
absorption cross section of the nitrate (Müller et al., 2014). For the MVK nitrate, no OH loss
correction was applied, because MVK nitrate is saturated and is not expected to undergo
significant loss to OH. However, its loss to wall uptake and photolysis loss was corrected,
following the same method as used for the ethanal nitrate. The MVK nitrate loss rates for wall
uptake and photolysis inside the chamber were set the same as those for the 4,1-carbonyl nitrate,
because MVK nitrate is also a ketone nitrate, which is prone to photolysis loss, and it has a
molecular weight close to that of the 4,1-carbonyl nitrate. The apparent yield is 24.5% for MVK





nitrate and 8.08% for ethanal nitrate. Considering the uncertainties in the sensitivities of MVK nitrate and ethanal nitrate (Table S2), the MVK nitrate yield is 24(±3)%, and the ethanal nitrate yield is 8(±2)%. The fractional inlet sampling loss for the three nitrates was determined by comparing the CIMS signals of sampling through the 5.2 m long 50°C tubing and through a 20 cm room temperature tubing. By correcting for the inlet sampling loss, the MVK nitrate yield is 24(±5)%, and the ethanal nitrate yield is 8(±3)%. For the two OH oxidation experiments, the first-order loss rate of the 4,1-carbonyl nitrate was $3 \times 10^{-4}\,s^{-1}$ (Fig. S8). Since the total wall uptake and photolysis loss rate for 4,1-isoprene carbonyl nitrate was $4.3 \times 10^{-5}\,s^{-1}$, approximately 85% of the 4,1-carbonyl nitrate was lost to OH oxidation. After correcting for this factor, the MVK nitrate yield is 28(±5)%, and the ethanal nitrate yield is 9(±3)%.

### 4.3.2 Photolysis

Previous work on acetaldehyde suggests that at 313 nm the dominant photolysis reaction is dissociation of the C−CHO bond, forming a formyl radical (•CHO) (Blacet and Loeffler, 1942). At shorter wavelength (265 nm), the reaction can proceed by intramolecular rearrangement forming $CH_4$ and CO (Blacet and Loeffler, 1942). For compounds with longer carbon chain length, such as propyl- and butyl- aldehydes, the photo-dissociation reaction can produce alkenes and smaller aldehydes at 238 nm and 187 nm (Blacet and Crane, 1954). Since the UV radiation that reaches the earth's surface is mostly above 300 nm, the formyl radical pathway is expected to be the most important photolysis reaction for alkyl aldehydes (Shepson and Heicklen, 1982). For the isoprene carbonyl nitrate, the C−CHO bond is strengthened by the delocalized electrons from the vinyl and the carbonyl groups, leading to a bond dissociation energy of 413 kJ/mol, as measured for acrolein, which is larger than the C−CHO bond dissociation energy of acetaldehyde (355 kJ/mol) (Wiberg et al., 1992). In comparison, the O−$NO_2$ bond dissociation energy is 175 kJ/mol (Luo, 2007b), much lower than the dissociation energy of the C−CHO bond. Hence, dissociation of the weak O−$NO_2$ bond may be an important reaction pathway for the carbonyl nitrate. This process likely involves the absorption of a photon by the C=C−C=O chromophore, followed by intramolecular energy redistribution to deposit energy into the O−$NO_2$ bond prior to dissociation. This reaction step would generate $NO_2$ and an alkoxy radical, which upon reaction with $O_2$ forms a conjugated dialdehyde.



Fig. 15 shows the CIMS spectra before and after the photolysis of the isoprene carbonyl nitrate.
Cyclohexane was used as the OH scavenger for this experiment. The CIMS signal for the
dialdehyde, which is the O−NO$_2$ bond dissociation product (reaction mechanism shown in Fig.
16), did not increase significantly. This may be because the CIMS was not sensitive to the
dialdehyde, and/or the dialdehyde underwent rapid secondary reactions, rendering its steady-state
concentration below the CIMS detection limit. Alternatively, it is possible that the alkoxy radical
derived from O−NO$_2$ bond dissociation undergoes a 1,5-H shift reaction (Fig. 16), rendering the
formation of the dialdehyde an insignificant pathway. The resulting alkyl radical can
immediately form a peroxy radical, which may follow the H shift mechanism proposed by
Peeters et al. (2009) and form a hydroperoxy aldehyde (HPALD) compound, as observed at m/z
257 by the CIMS (Fig. 15). When the peroxy radical reacts with NO or RO$_2$, the resulting alkoxy
radical will form a hydroxy dialdehyde (Fig. 16) with m/z ratio at 241, which was also observed
by the CIMS (Fig. 14). It is worth noting that we also observed CIMS signals for the
deprotonated ions derived from the HPALD compound (m/z 129 and m/z 147) and the hydroxy
dialdehyde (m/z 113 and m/z 131). The proton transfer reaction between the iodide ion and
alcohols/peroxides have not been observed previously, but it is possible that the conjugated
structures help stabilize the charge and hence make the proton transfer reaction a viable reaction
channel.
The product at m/z 276 has the molecular weight of MVK nitrate. In the presence of OH
scavenger, however, the reaction is unlikely to proceed by the OH-initiated oxidation pathway to
form MVK nitrate. Instead, we hypothesize that the isoprene carbonyl nitrate could dissociate via
the C−CHO bond, which, following reaction with O$_2$ and HO$_2$, would form a vinyl
hydroperoxide with the same molecular weight as MVK nitrate. Vinyl hydroperoxides are
known to be a reactive intermediate from the intramolecular H shift of Criegee biradical, which
can decompose into OH and alkoxy radicals (Kroll et al., 2002). However, the un-energized
vinyl hydroperoxides should have a lifetime long enough to be detected by mass spectrometers
(Liu et al., 2015). In fact, theoretical calculations suggest that at 25 °C vinyl hydroperoxide has a
lifetime of 58 hours (Richardson, 1995). Therefore, the product at m/z 276 is likely the vinyl
hydroperoxide. For the OH oxidation product experiments, however, we attributed m/z 276 to
MVK nitrate only, because RO$_2$ + NO reaction (forming MVK nitrate) should dominate over
RO$_2$ + HO$_2$ reaction (forming vinyl hydroperoxide), in the presence of high NO concentration.





Based on the CIMS spectra of the photolysis products, we conclude that the photolysis of the
isoprene carbonyl nitrate leads to the dissociation of both the O−NO$_2$ and the C−CHO bonds. A
reaction scheme is proposed in Fig. 16. Future studies are needed to evaluate the relative
importance of these two processes.
**5  Conclusions and future work**
An isoprene-derived carbonyl nitrate model compound was synthesized to study its
photochemical degradation chemistry in the atmosphere. The UV absorption spectrum of this
compound has contributions from both the C=C−C=O and the –ONO$_2$ chromophores, as is
confirmed by theoretical calculations, but absorption in the actinic region involves a transition
involving the carbonyl group. The combination of the C=C−C=O and the –ONO$_2$ chromophores
enhances the UV cross section of this molecule relative to alkyl nitrates, making photolysis its
dominant daytime sink. The photochemical lifetime of the carbonyl nitrate can be less than one
hour, due to its rapid photolysis loss, together with high reactivity toward OH and O$_3$. The OH
and O$_3$ oxidation rate constants for the 4,1-isoprene carbonyl nitrate obtained in this study were
both smaller than the reported rate constants for the δ-isoprene hydroxy nitrates (Jacobs et al.,
2014; Lee et al., 2014b). This could be because the oxidation by either OH or O$_3$ would break the
resonance structure of the C=C−C=O moiety, thus increasing the activation energy.
Using the iodide-based CIMS, we identified the first-generation nitrate products from the OH-
initiated oxidation of the synthesized carbonyl nitrate, including mononitrate, dinitrate and
nitrooxy peroxyacyl nitrate. Two of the products, the MVK nitrate and the ethanal nitrate, were
quantified, which contributed to 37(±5)% of the total products. The CIMS spectra of the nitrate
photolysis products suggest that both the C−CHO bond and the O−NO$_2$ bond dissociate in the
reaction. Since photolysis is a significant sink for the carbonyl nitrate, it is important for future
studies to investigate the relative importance of the two reaction pathways, in order to fully
understand the fate of NO$_x$ in isoprene-rich atmospheres. Dissociation of the O−NO$_2$ bond may
afford highly oxidized alcohol and hydroperoxide, which can potentially undergo uptake into the
particle phase and facilitate the formation of secondary organic aerosols. The C−CHO
dissociation pathway may form a vinyl hydroperoxide product.



The NO$_3$-initiated isoprene oxidation can produce a series of carbonyl nitrates. The 1,4-carbonyl
nitrate, which is the dominant isomer, is expected to have similar photolysis reactivity as the 4,1-
carbonyl nitrate studied in this work, because they both have the O=C−C=C−C chromophore and
the −ONO$_2$ chromophore, which would enhance the molecular absorption cross section. The
influence of the unsaturated ketone functionality on nitrate photolysis is still unclear, and future
studies are needed to understand how the different conjugated structures can affect the
photochemical processes.
The experiments in this work were conducted in the presence of relatively high NO
concentration. In the ambient environment, organic nitrates produced in the high NO$_x$ regime can
undergo photochemical degradation in the low NO regime, due to the wide span of ambient NO$_x$
concentrations (Su et al., 2015; Xiong et al., 2015). Crounse et al. (2012) proposed that under
low NO conditions, the oxidation of methacrolein (MACR) can regenerate OH radicals and form
a lactone that is prone to reactive uptake onto the aerosol phase. Since the 4,1-carbonyl nitrate
has a structure similar to that of MACR, it might also undergo similar reaction in the clean
environment. Further experimental work is needed to investigate how the photochemical
oxidation process of the carbonyl nitrate can influence the formation of OH radicals and growth
of secondary organic aerosols.
**Acknowledgement**
This research was supported in part through computational resources provided by Information
Technology at Purdue University. We thank the National Science Foundation for supporting
CHB and LVS (grant CHE-1465154), and FX and PBS (grant AGS-1228496).

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

Supplementary functions for Gaussian basis sets, The Journal of Chemical Physics, 80, 3265-
3269, doi:http://dx.doi.org/10.1063/1.447079, 1984.
Giacopelli, P., Ford, K., Espada, C., and Shepson, P. B.: Comparison of the measured and
simulated isoprene nitrate distributions above a forest canopy, Journal of Geophysical Research,
110, D01304, 10.1029/2004jd005123, 2005.
Gilbert, A. T. B.: IQmol molecular viewer, 2012.
Gray, G. M.: Method for the preparation of (E)-4-bromo-2-methylbut-2-en-1-al 4288635, 1981.
Grosjean, D., Grosjean, E., and Williams, E. L.: Thermal decomposition of C3-substituted
peroxyacyl nitrates, Res Chem Intermed, 20, 447-461, 10.1163/156856794X00414, 1994.
Grossenbacher, J. W., Barket Jr, D. J., Shepson, P. B., Carroll, M. A., Olszyna, K., and Apel, E.: A
comparison of isoprene nitrate concentrations at two forest-impacted sites, Journal of
Geophysical Research: Atmospheres, 109, D11311, 10.1029/2003JD003966, 2004.
Haagen-Smit, A. J.: Chemistry and Physiology of Los Angeles Smog, Industrial & Engineering
Chemistry, 44, 1342-1346, 10.1021/ie50510a045, 1952.
Hallquist, M., WÄngberg, I., and LjungstrÖm, E.: Atmospheric Fate of Carbonyl Oxidation
Products Originating from α-Pinene and Δ3-Carene:  Determination of Rate of Reaction with OH



and NO3 Radicals, UV Absorption Cross Sections, and Vapor Pressures, Environmental science &
technology, 31, 3166-3172, 10.1021/es970151a, 1997.
Herron, J. T., and Huie, R. E.: Rate constants for the reactions of ozone with ethene and
propene, from 235.0 to 362.0.deg.K, The Journal of Physical Chemistry, 78, 2085-2088,
10.1021/j100614a004, 1974.
Hohenberg, P., and Kohn, W.: Inhomogeneous Electron Gas, Physical Review, 136, B864-B871,
594    1964.

Horowitz, L. W., Fiore, A. M., Milly, G. P., Cohen, R. C., Perring, A., Wooldridge, P. J., Hess, P. G.,
Emmons, L. K., and Lamarque, J.-F.: Observational constraints on the chemistry of isoprene
nitrates over the eastern United States, Journal of Geophysical Research, 112, D12S08,
10.1029/2006jd007747, 2007.
Hu, K. S., Darer, A. I., and Elrod, M. J.: Thermodynamics and kinetics of the hydrolysis of
atmospherically relevant organonitrates and organosulfates, Atmospheric Chemistry and
Physics, 11, 8307-8320, 10.5194/acp-11-8307-2011, 2011.
Jacobs, M. I., Burke, W. J., and Elrod, M. J.: Kinetics of the reactions of isoprene-derived
hydroxynitrates: gas phase epoxide formation and solution phase hydrolysis, Atmospheric
Chemistry and Physics, 14, 8933-8946, 10.5194/acp-14-8933-2014, 2014.
Kabir, M., Jagiella, S., and Zabel, F.: Thermal Stability of n-Acyl Peroxynitrates, International
Journal of Chemical Kinetics, 46, 462-469, 10.1002/kin.20862, 2014.
Klein, T., Barnes, I., Becker, K. H., Fink, E. H., and Zabel, F.: Pressure dependence of the rate
constants for the reactions of ethene and propene with hydroxyl radicals at 295 K, The Journal
of Physical Chemistry, 88, 5020-5025, 10.1021/j150665a046, 1984.
Kohn, W., and Sham, L. J.: Self-Consistent Equations Including Exchange and Correlation Effects,
Physical Review, 140, A1133-A1138, 1965.
Kroll, J. H., Donahue, N. M., Cee, V. J., Demerjian, K. L., and Anderson, J. G.: Gas-Phase
Ozonolysis of Alkenes: Formation of OH from Anti Carbonyl Oxides, Journal of the American
Chemical Society, 124, 8518-8519, 10.1021/ja0266060, 2002.
Kwan, A. J., Chan, A. W. H., Ng, N. L., Kjaergaard, H. G., Seinfeld, J. H., and Wennberg, P. O.:
Peroxy radical chemistry and OH radical production during the NO$_3$-initiated
oxidation of isoprene, Atmospheric Chemistry and Physics, 12, 7499-7515, 10.5194/acp-12-
618    7499-2012, 2012.

Kwok, E. S. C., and Atkinson, R.: Estimation of hydroxyl radical reaction rate constants for gas-
phase organic compounds using a structure-reactivity relationship: An update, Atmospheric
Environment, 29, 1685-1695, http://dx.doi.org/10.1016/1352-2310(95)00069-B, 1995.



Lee, B. H., Lopez-Hilfiker, F. D., Mohr, C., Kurtén, T., Worsnop, D. R., and Thornton, J. A.: An Iodide-Adduct High-Resolution Time-of-Flight Chemical-Ionization Mass Spectrometer: Application to Atmospheric Inorganic and Organic Compounds, Environmental science & technology, 48, 6309-6317, 10.1021/es500362a, 2014a.

Lee, L., Teng, A. P., Wennberg, P. O., Crounse, J. D., and Cohen, R. C.: On rates and mechanisms of OH and O3 reactions with isoprene-derived hydroxy nitrates, The journal of physical chemistry. A, 118, 1622-1637, 10.1021/jp4107603, 2014b.

Lefohn, A. S., and Foley, J. K.: Establishing Relevant Ozone Standards to Protect Vegetation and Human Health: Exposure/Dose-Response Considerations, Air & Waste, 43, 106-112, 10.1080/1073161X.1993.10467111, 1993.

Linder, B., and Abdulnur, S.: Solvent Effects on Electronic Spectral Intensities, The Journal of Chemical Physics, 54, 1807-1814, doi:http://dx.doi.org/10.1063/1.1675088, 1971.

Lippmann, M.: HEALTH EFFECTS OF OZONE A Critical Review, JAPCA, 39, 672-695, 10.1080/08940630.1989.10466554, 1989.

Liu, F., Fang, Y., Kumar, M., Thompson, W. H., and Lester, M. I.: Direct observation of vinyl hydroperoxide, Physical Chemistry Chemical Physics, 17, 20490-20494, 10.1039/C5CP02917A, 2015.

Lockwood, A. L., Shepson, P. B., Fiddler, M. N., and Alaghmand, M.: Isoprene nitrates: preparation, separation, identification, yields, and atmospheric chemistry, Atmospheric Chemistry and Physics, 10, 6169-6178, 10.5194/acp-10-6169-2010, 2010.

Lu, K. D., Rohrer, F., Holland, F., Fuchs, H., Bohn, B., Brauers, T., Chang, C. C., Häseler, R., Hu, M., Kita, K., Kondo, Y., Li, X., Lou, S. R., Nehr, S., Shao, M., Zeng, L. M., Wahner, A., Zhang, Y. H., and Hofzumahaus, A.: Observation and modelling of OH and HO2 concentrations in the Pearl River Delta 2006: a missing OH source in a VOC rich atmosphere, Atmos. Chem. Phys., 12, 1541-1569, 10.5194/acp-12-1541-2012, 2012.

Luo, Y.-R.: BDEs in the halogenated molecules, clusters and complexes, in: Comprehensive Handbook of Chemical Bond Energies, CRC Press, 1351-1427, 2007a.

Luo, Y.-R.: BDEs of O-X bonds, in: Comprehensive Handbook of Chemical Bond Energies, CRC Press, 351, 2007b.

Madronich, S., and Flocke, S.: The role of solar radiation in atmospheric chemistry, in: Handbook of Environmental Chemistry, edited by: Boule, P., Springer-Verlag, Heidelberg, 1-26, 1998.

Mao, J., Paulot, F., Jacob, D. J., Cohen, R. C., Crounse, J. D., Wennberg, P. O., Keller, C. A., Hudman, R. C., Barkley, M. P., and Horowitz, L. W.: Ozone and organic nitrates over the eastern



United States: Sensitivity to isoprene chemistry, Journal of Geophysical Research: Atmospheres, 118, 11,256-211,268, 10.1002/jgrd.50817, 2013.

Martinez, M., Harder, H., Kovacs, T. A., Simpas, J. B., Bassis, J., Lesher, R., Brune, W. H., Frost, G. J., Williams, E. J., Stroud, C. A., Jobson, B. T., Roberts, J. M., Hall, S. R., Shetter, R. E., Wert, B., Fried, A., Alicke, B., Stutz, J., Young, V. L., White, A. B., and Zamora, R. J.: OH and HO2 concentrations, sources, and loss rates during the Southern Oxidants Study in Nashville, Tennessee, summer 1999, Journal of Geophysical Research: Atmospheres, 108, n/a-n/a, 10.1029/2003JD003551, 2003.

Mihelcic, D., Holland, F., Hofzumahaus, A., Hoppe, L., Konrad, S., Müsgen, P., Pätz, H. W., Schäfer, H. J., Schmitz, T., Volz-Thomas, A., Bächmann, K., Schlomski, S., Platt, U., Geyer, A., Alicke, B., and Moortgat, G. K.: Peroxy radicals during BERLIOZ at Pabstthum: Measurements, radical budgets and ozone production, Journal of Geophysical Research: Atmospheres, 108, n/a-n/a, 10.1029/2001JD001014, 2003.

Müller, J. F., Peeters, J., and Stavrakou, T.: Fast photolysis of carbonyl nitrates from isoprene, Atmospheric Chemistry and Physics, 14, 2497-2508, 10.5194/acp-14-2497-2014, 2014.

Neuman, J. A., Nowak, J. B., Huey, L. G., Burkholder, J. B., Dibb, J. E., Holloway, J. S., Liao, J., Peischl, J., Roberts, J. M., Ryerson, T. B., Scheuer, E., Stark, H., Stickel, R. E., Tanner, D. J., and Weinheimer, A.: Bromine measurements in ozone depleted air over the Arctic Ocean, Atmos. Chem. Phys., 10, 6503-6514, 10.5194/acp-10-6503-2010, 2010.

Noyes, W. A.: Explanation of the Formation of Alkyl Nitrites in Dilute Solutions; Butyl and Amyl Nitrites, Journal of the American Chemical Society, 55, 3888-3889, 10.1021/ja01336a503, 1933.

Parrish, D. D., Lamarque, J. F., Naik, V., Horowitz, L., Shindell, D. T., Staehelin, J., Derwent, R., Cooper, O. R., Tanimoto, H., Volz-Thomas, A., Gilge, S., Scheel, H. E., Steinbacher, M., and Fröhlich, M.: Long-term changes in lower tropospheric baseline ozone concentrations: Comparing chemistry-climate models and observations at northern midlatitudes, Journal of Geophysical Research: Atmospheres, 119, 5719-5736, 10.1002/2013JD021435, 2014.

Patchen, A. K., Pennino, M. J., Kiep, A. C., and Elrod, M. J.: Direct kinetics study of the product-forming channels of the reaction of isoprene-derived hydroxyperoxy radicals with NO, International Journal of Chemical Kinetics, 39, 353-361, 10.1002/kin.20248, 2007.

Paulot, F., Crounse, J. D., Kjaergaard, H. G., Kroll, J. H., Seinfeld, J. H., and Wennberg, P. O.: Isoprene photooxidation: new insights into the production of acids and organic nitrates, Atmos. Chem. Phys., 9, 1479-1501, 10.5194/acp-9-1479-2009, 2009.

Paulot, F., Henze, D. K., and Wennberg, P. O.: Impact of the isoprene photochemical cascade on tropical ozone, Atmospheric Chemistry and Physics, 12, 1307-1325, 10.5194/acp-12-1307-2012, 2012.

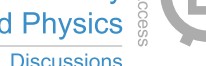
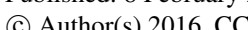


Peeters, J., Boullart, W., Pultau, V., Vandenberk, S., and Vereecken, L.: Structure–Activity
Relationship for the Addition of OH to (Poly)alkenes: Site-Specific and Total Rate Constants,
The Journal of Physical Chemistry A, 111, 1618-1631, 10.1021/jp066973o, 2007.
Peeters, J., Nguyen, T. L., and Vereecken, L.: HOx radical regeneration in the oxidation of
isoprene, Physical chemistry chemical physics : PCCP, 11, 5935-5939, 10.1039/b908511d, 2009.
Perring, A. E., Wisthaler, A., Graus, M., Wooldridge, P. J., Lockwood, A. L., Mielke, L. H., Shepson,
P. B., Hansel, A., and Cohen, R. C.: A product study of the isoprene+NO3 reaction, Atmos. Chem.
Phys., 9, 4945-4956, 10.5194/acp-9-4945-2009, 2009.
Platt, U., Alicke, B., Dubois, R., Geyer, A., Hofzumahaus, A., Holland, F., Martinez, M., Mihelcic,
D., Klüpfel, T., Lohrmann, B., Pätz, W., Perner, D., Rohrer, F., Schäfer, J., and Stutz, J.: Free
Radicals and Fast Photochemistry during BERLIOZ, Journal of Atmospheric Chemistry, 42, 359-
394, 10.1023/A:1015707531660, 2002.
Richardson, W. H.: An Evaluation of Vinyl Hydroperoxide as an Isolable Molecule, The Journal of
Organic Chemistry, 60, 4090-4095, 10.1021/jo00118a027, 1995.
Rindelaub, J. D., McAvey, K. M., and Shepson, P. B.: The photochemical production of organic
nitrates from α-pinene and loss via acid-dependent particle phase hydrolysis, Atmospheric
Environment, 100, 193-201, http://dx.doi.org/10.1016/j.atmosenv.2014.11.010, 2015.
Roberts, J. M.: The atmospheric chemistry of organic nitrates, Atmospheric Environment. Part A.
General Topics, 24, 243-287, http://dx.doi.org/10.1016/0960-1686(90)90108-Y, 1990.
Roberts, J. M., and Bertman, S. B.: The thermal decomposition of peroxyacetic nitric anhydride
(PAN) and peroxymethacrylic nitric anhydride (MPAN), International Journal of Chemical
Kinetics, 24, 297-307, 10.1002/kin.550240307, 1992.
Roberts, J. M., Flocke, F., Stroud, C. A., Hereid, D., Williams, E., Fehsenfeld, F., Brune, W.,
Martinez, M., and Harder, H.: Ground-based measurements of peroxycarboxylic nitric
anhydrides (PANs) during the 1999 Southern Oxidants Study Nashville Intensive, Journal of
Geophysical Research: Atmospheres, 107, ACH 1-1-ACH 1-10, 10.1029/2001JD000947, 2002.
Rollins, A. W., Kiendler-Scharr, A., Fry, J. L., Brauers, T., Brown, S. S., Dorn, H. P., Dubé, W. P.,
Fuchs, H., Mensah, A., Mentel, T. F., Rohrer, F., Tillmann, R., Wegener, R., Wooldridge, P. J., and
Cohen, R. C.: Isoprene oxidation by nitrate radical: alkyl nitrate and secondary organic aerosol
yields, Atmos. Chem. Phys., 9, 6685-6703, 10.5194/acp-9-6685-2009, 2009.
Runge, E., and Gross, E. K. U.: Density-Functional Theory for Time-Dependent Systems, Physical
Review Letters, 52, 997-1000, 1984.
Schwantes, R. H., Teng, A. P., Nguyen, T. B., Coggon, M. M., Crounse, J. D., St. Clair, J. M., Zhang,
X., Schilling, K. A., Seinfeld, J. H., and Wennberg, P. O.: Isoprene NO3 Oxidation Products from
the RO2 + HO2 Pathway, The Journal of Physical Chemistry A, 10.1021/acs.jpca.5b06355, 2015.



Shao, Y., Gan, Z., Epifanovsky, E., Gilbert, A. T. B., Wormit, M., Kussmann, J., Lange, A. W., Behn,
A., Deng, J., Feng, X., Ghosh, D., Goldey, M., Horn, P. R., Jacobson, L. D., Kaliman, I., Khaliullin, R.
Z., Kuś, T., Landau, A., Liu, J., Proynov, E. I., Rhee, Y. M., Richard, R. M., Rohrdanz, M. A., Steele,
R. P., Sundstrom, E. J., Woodcock, H. L., Zimmerman, P. M., Zuev, D., Albrecht, B., Alguire, E.,
Austin, B., Beran, G. J. O., Bernard, Y. A., Berquist, E., Brandhorst, K., Bravaya, K. B., Brown, S. T.,
Casanova, D., Chang, C.-M., Chen, Y., Chien, S. H., Closser, K. D., Crittenden, D. L., Diedenhofen,
M., DiStasio, R. A., Do, H., Dutoi, A. D., Edgar, R. G., Fatehi, S., Fusti-Molnar, L., Ghysels, A.,
Golubeva-Zadorozhnaya, A., Gomes, J., Hanson-Heine, M. W. D., Harbach, P. H. P., Hauser, A.
W., Hohenstein, E. G., Holden, Z. C., Jagau, T.-C., Ji, H., Kaduk, B., Khistyaev, K., Kim, J., Kim, J.,
King, R. A., Klunzinger, P., Kosenkov, D., Kowalczyk, T., Krauter, C. M., Lao, K. U., Laurent, A. D.,
Lawler, K. V., Levchenko, S. V., Lin, C. Y., Liu, F., Livshits, E., Lochan, R. C., Luenser, A., Manohar,
P., Manzer, S. F., Mao, S.-P., Mardirossian, N., Marenich, A. V., Maurer, S. A., Mayhall, N. J.,
Neuscamman, E., Oana, C. M., Olivares-Amaya, R., O'Neill, D. P., Parkhill, J. A., Perrine, T. M.,
Peverati, R., Prociuk, A., Rehn, D. R., Rosta, E., Russ, N. J., Sharada, S. M., Sharma, S., Small, D.
W., Sodt, A., Stein, T., Stück, D., Su, Y.-C., Thom, A. J. W., Tsuchimochi, T., Vanovschi, V., Vogt, L.,
Vydrov, O., Wang, T., Watson, M. A., Wenzel, J., White, A., Williams, C. F., Yang, J., Yeganeh, S.,
Yost, S. R., You, Z.-Q., Zhang, I. Y., Zhang, X., Zhao, Y., Brooks, B. R., Chan, G. K. L., Chipman, D.
M., Cramer, C. J., Goddard, W. A., Gordon, M. S., Hehre, W. J., Klamt, A., Schaefer, H. F.,
Schmidt, M. W., Sherrill, C. D., Truhlar, D. G., Warshel, A., Xu, X., Aspuru-Guzik, A., Baer, R., Bell,
A. T., Besley, N. A., Chai, J.-D., Dreuw, A., Dunietz, B. D., Furlani, T. R., Gwaltney, S. R., Hsu, C.-P.,
Jung, Y., Kong, J., Lambrecht, D. S., Liang, W., Ochsenfeld, C., Rassolov, V. A., Slipchenko, L. V.,
Subotnik, J. E., Van Voorhis, T., Herbert, J. M., Krylov, A. I., Gill, P. M. W., and Head-Gordon, M.:
Advances in molecular quantum chemistry contained in the Q-Chem 4 program package,
Molecular Physics, 113, 184-215, 10.1080/00268976.2014.952696, 2015.
Shepson, P. B., and Heicklen, J.: The wavelength and pressure dependence of the photolysis of
propionaldehyde in air, Journal of Photochemistry, 19, 215-227,
http://dx.doi.org/10.1016/0047-2670(82)80024-5, 1982.
Shepson, P. B., Bottenheim, J. W., Hastie, D. R., and Venkatram, A.: Determination of the
relative ozone and PAN deposition velocities at night, Geophysical Research Letters, 19, 1121-
1124, 10.1029/92GL01118, 1992.
Sprengnether, M., Demerjian, K. L., Donahue, N. M., and Anderson, J. G.: Product analysis of the
OH oxidation of isoprene and 1,3-butadiene in the presence of NO, Journal of Geophysical
Research, 107, 10.1029/2001jd000716, 2002.
Starn, T. K., Shepson, P. B., Bertman, S. B., Riemer, D. D., Zika, R. G., and Olszyna, K.: Nighttime
isoprene chemistry at an urban-impacted forest site, Journal of Geophysical Research:
Atmospheres, 103, 22437-22447, 10.1029/98JD01201, 1998.
Su, L., Patton, E. G., Vilà-Guerau de Arellano, J., Guenther, A. B., Kaser, L., Yuan, B., Xiong, F.,
Shepson, P. B., Zhang, L., Miller, D. O., Brune, W. H., Baumann, K., Edgerton, E., Weinheimer, A.,
and Mak, J. E.: Understanding isoprene photo-oxidation using observations and modelling over



a subtropical forest in the Southeast US, Atmos. Chem. Phys. Discuss., 15, 31621-31663,
10.5194/acpd-15-31621-2015, 2015.
Suarez-Bertoa, R., Picquet-Varrault, B., Tamas, W., Pangui, E., and Doussin, J. F.: Atmospheric
fate of a series of carbonyl nitrates: photolysis frequencies and OH-oxidation rate constants,
Environmental science & technology, 46, 12502-12509, 10.1021/es302613x, 2012.
Treacy, J., Hag, M. E., O'Farrell, D., and Sidebottom, H.: Reactions of Ozone with Unsaturated
Organic Compounds, Berichte der Bunsengesellschaft für physikalische Chemie, 96, 422-427,
10.1002/bbpc.19920960337, 1992.
Tuazon, E. C., Atkinson, R., Mac Leod, H., Biermann, H. W., Winer, A. M., Carter, W. P. L., and
Pitts, J. N.: Yields of glyoxal and methylglyoxal from the nitrogen oxide(NOx)-air
photooxidations of toluene and m- and p-xylene, Environmental science & technology, 18, 981-
984, 10.1021/es00130a017, 1984.
Tuazon, E. C., and Atkinson, R.: A product study of the gas-phase reaction of Isoprene with the
OH radical in the presence of NOx, International Journal of Chemical Kinetics, 22, 1221-1236,
10.1002/kin.550221202, 1990.
Vingarzan, R.: A review of surface ozone background levels and trends, Atmospheric
Environment, 38, 3431-3442, http://dx.doi.org/10.1016/j.atmosenv.2004.03.030, 2004.
Wang, X., Wang, T., Yan, C., Tham, Y. J., Xue, L., Xu, Z., and Zha, Q.: Large daytime signals of
N2O5 and NO3 inferred at 62 amu in a TD-CIMS: chemical interference or a real atmospheric
phenomenon?, Atmos. Meas. Tech., 7, 1-12, 10.5194/amt-7-1-2014, 2014.
Wiberg, K. B., Hadad, C. M., Rablen, P. R., and Cioslowski, J.: Substituent effects. 4. Nature of
substituent effects at carbonyl groups, Journal of the American Chemical Society, 114, 8644-
8654, 10.1021/ja00048a044, 1992.
Wu, S., Mickley, L. J., Jacob, D. J., Logan, J. A., Yantosca, R. M., and Rind, D.: Why are there large
differences between models in global budgets of tropospheric ozone?, Journal of Geophysical
Research: Atmospheres, 112, D05302, 10.1029/2006JD007801, 2007.
Xie, Y., Paulot, F., Carter, W. P. L., Nolte, C. G., Luecken, D. J., Hutzell, W. T., Wennberg, P. O.,
Cohen, R. C., and Pinder, R. W.: Understanding the impact of recent advances in isoprene
photooxidation on simulations of regional air quality, Atmospheric Chemistry and Physics, 13,
8439-8455, 10.5194/acp-13-8439-2013, 2013.
Xiong, F., McAvey, K. M., Pratt, K. A., Groff, C. J., Hostetler, M. A., Lipton, M. A., Starn, T. K.,
Seeley, J. V., Bertman, S. B., Teng, A. P., Crounse, J. D., Nguyen, T. B., Wennberg, P. O., Misztal,
P. K., Goldstein, A. H., Guenther, A. B., Koss, A. R., Olson, K. F., de Gouw, J. A., Baumann, K.,
Edgerton, E. S., Feiner, P. A., Zhang, L., Miller, D. O., Brune, W. H., and Shepson, P. B.:
Observation of isoprene hydroxynitrates in the southeastern United States and implications for
the fate of NOx, Atmos. Chem. Phys., 15, 11257-11272, 10.5194/acp-15-11257-2015, 2015.





Zellner, R., and Lorenz, K.: Laser photolysis/resonance fluorescence study of the rate constants
for the reactions of hydroxyl radicals with ethene and propene, The Journal of Physical
Chemistry, 88, 984-989, 10.1021/j150649a028, 1984.

**List of Figures**





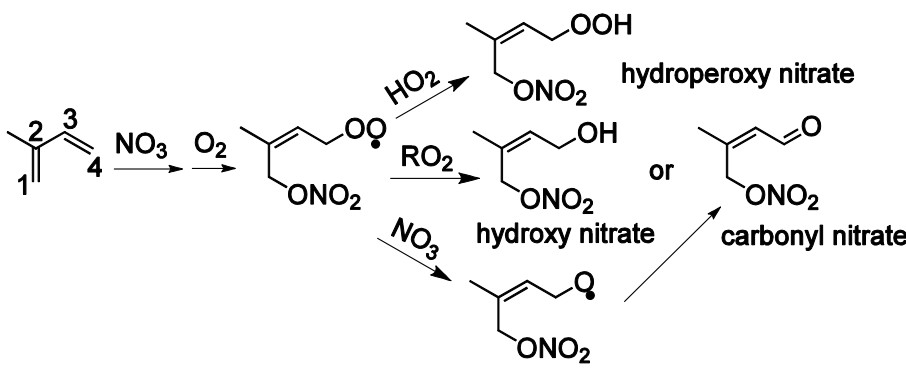


846 Figure 1. Organic nitrates produced from NO₃-initiated isoprene oxidation.





851 Figure 2. The synthesis route for the 4,1-isoprene carbonyl nitrate.






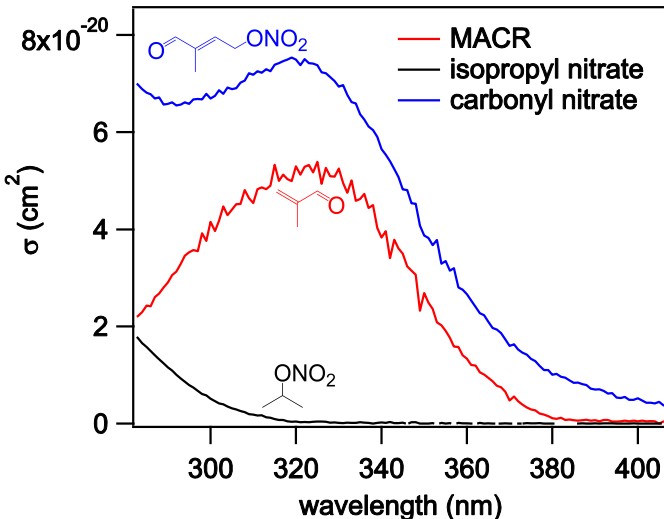


Figure 3. UV absorption cross section of the carbonyl nitrate, MACR and isoproyl nitrate. The
spectra were obtained in acetonitrile solvent.



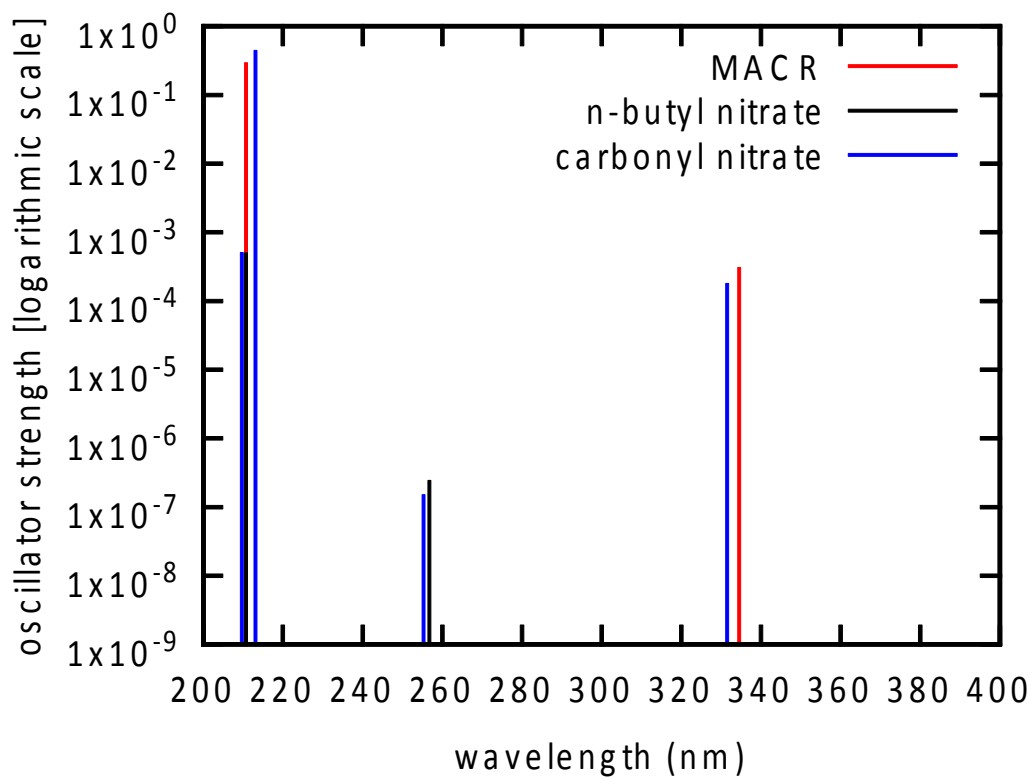


Figure 4. Theoretical absorption spectra of the carbonyl nitrate, MACR, and *n*-butyl nitrate in the
gas phase.




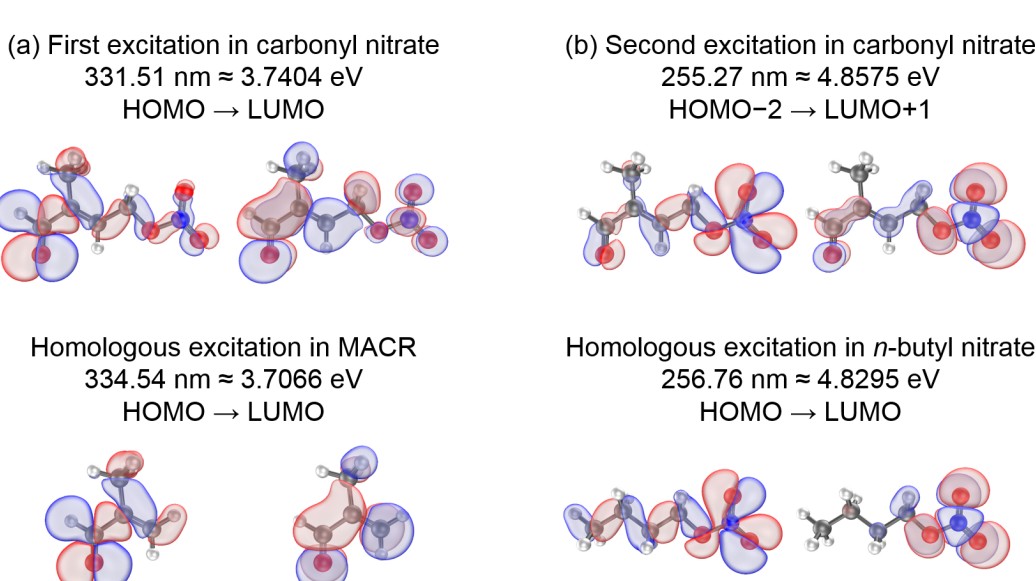


Figure 5. Molecular orbital analysis of the first (a) and second (b) electronic excitation of the
carbonyl nitrate. The blue and red colors represent the opposite phases of molecular orbitals.






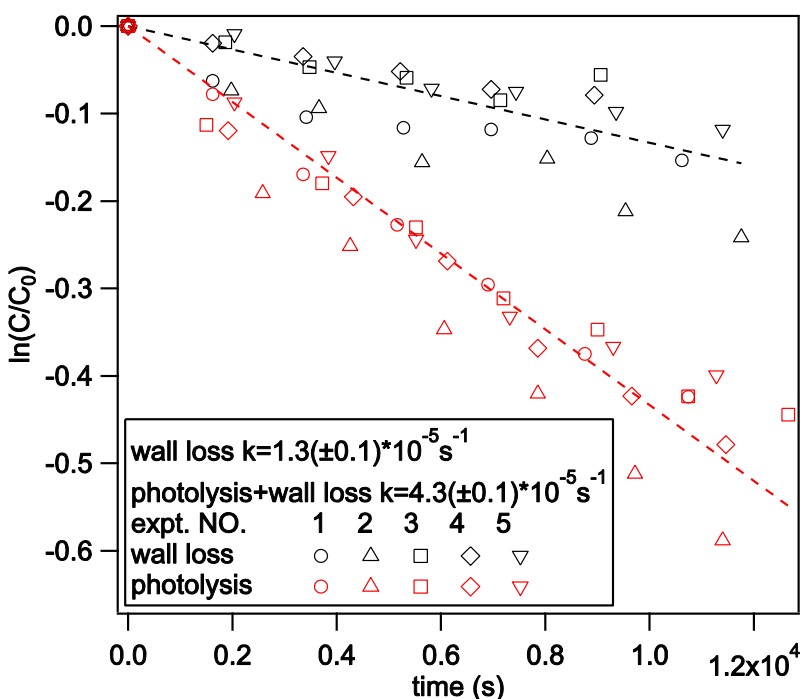


Figure 6. Wall loss and photolysis loss of the carbonly nitrate in the reaction chamber.




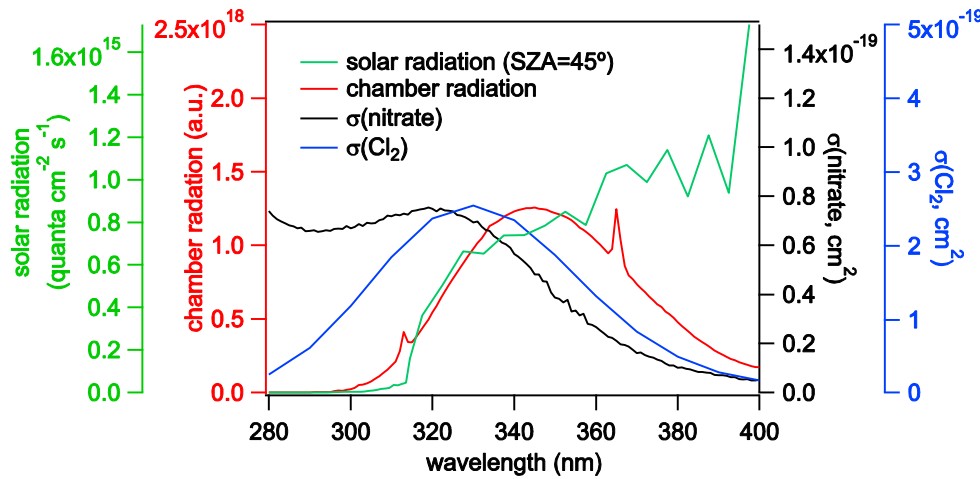


Figure 7. The radiation spectra of the chamber (red) and the sun (green, SZA=45° as an example),

and the absorption spectra of the carbonyl nitrate (black) and chlorine (blue).



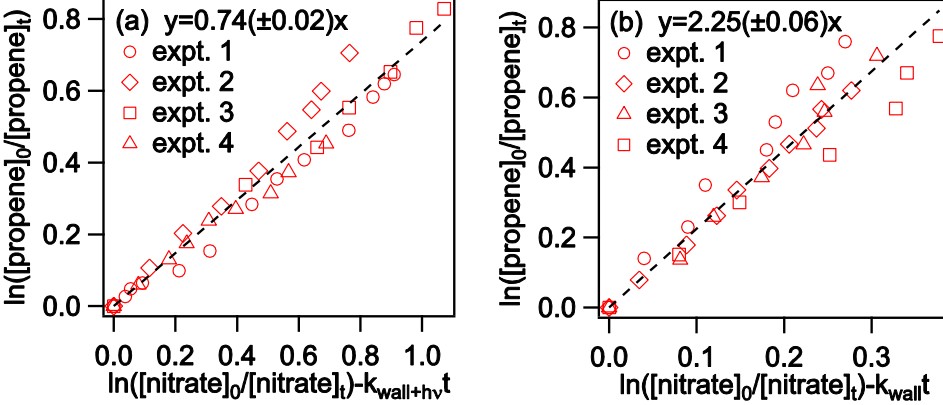


Figure 8. The first-order loss of propene relative to that of the carbonyl nitrate for OH-initiated (a)

and $O_3$-initiated (b) oxidation reactions.









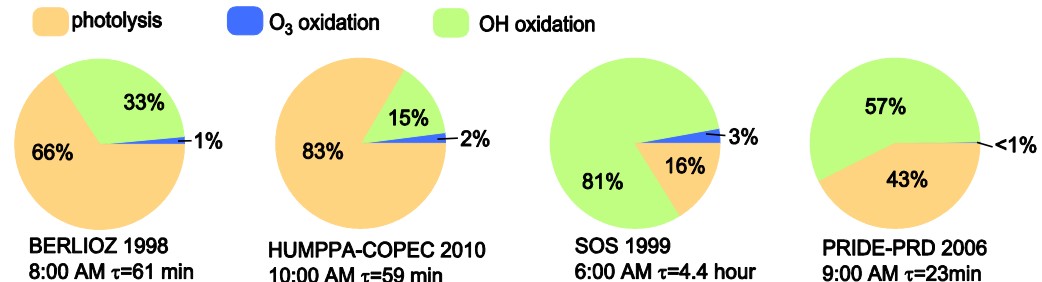


Figure 9. The relative contributions of photolysis (orange), OH oxidation (green) and $O_3$ oxidation (blue) to the photochemical decay of the carbonyl nitrate, calculated based on measured OH and $O_3$ concentrations for the following field studies: BERLIOZ 1998 study at Pabstthum, Germany (Platt et al., 2002; Mihelcic et al., 2003), HUMPPA-COPEC 2010 study at Hyytiälä, Finland, SOS 1999 study at Nashville, US (Roberts et al., 2002; Martinez et al., 2003) and PRIDE-PRD 2006 study at Guangzhou, China (Lu et al., 2012).



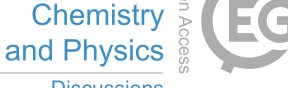



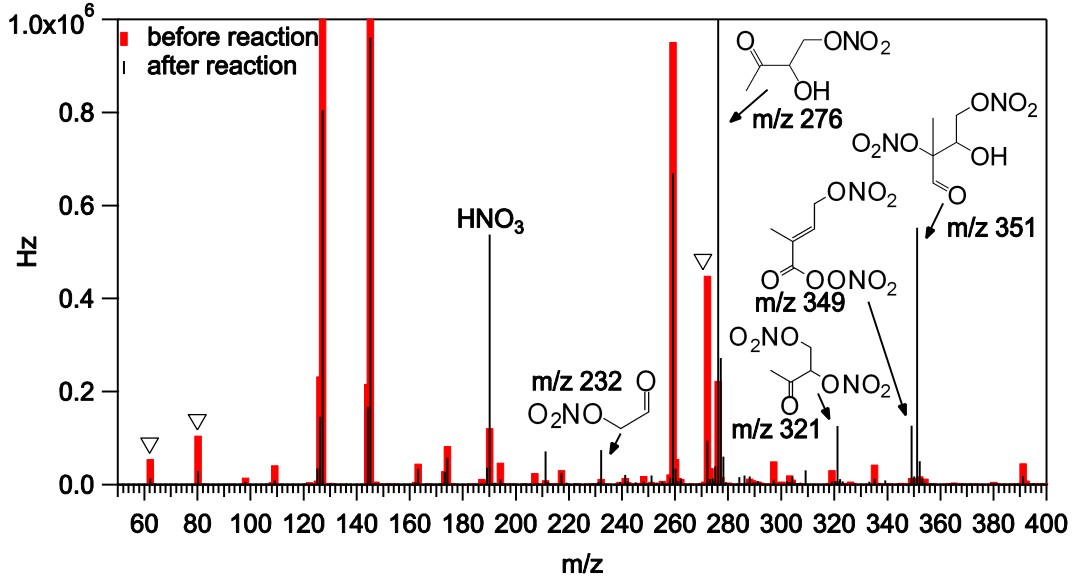


Figure 10. The CIMS spectra before (red) and after (black) the OH + carbonyl nitrate oxidation
reaction.  The inverted triangles show the decreases in CIMS signals for the carbonyl nitrate (m/z
272) and the $NO_3^-$ fragments (m/z 62, water cluster at m/z 80) derived from the carbonyl nitrate
(Fig. 11).





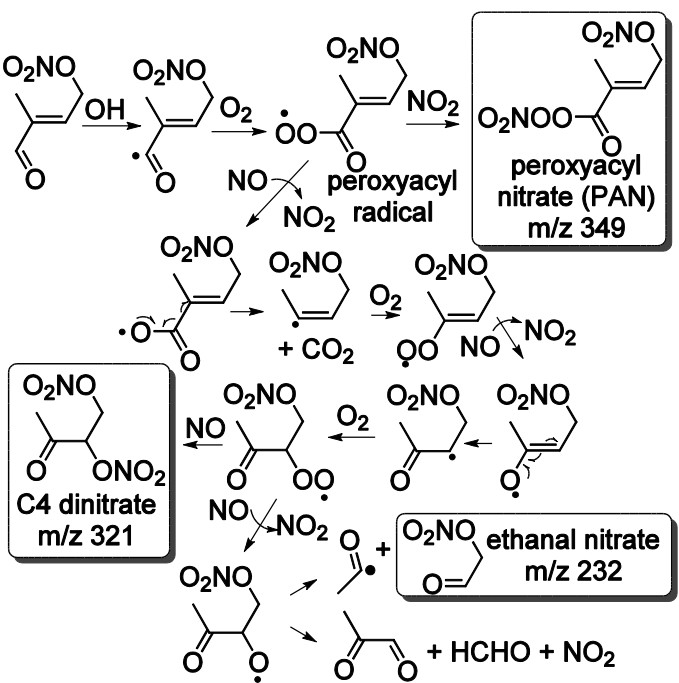


Figure 11. A proposed reaction mechanism for the H abstraction pathway for the OH + carbonyl
nitrate oxidation reaction. The compounds in boxes are proposed products as observed by the
CIMS (Fig. 8).





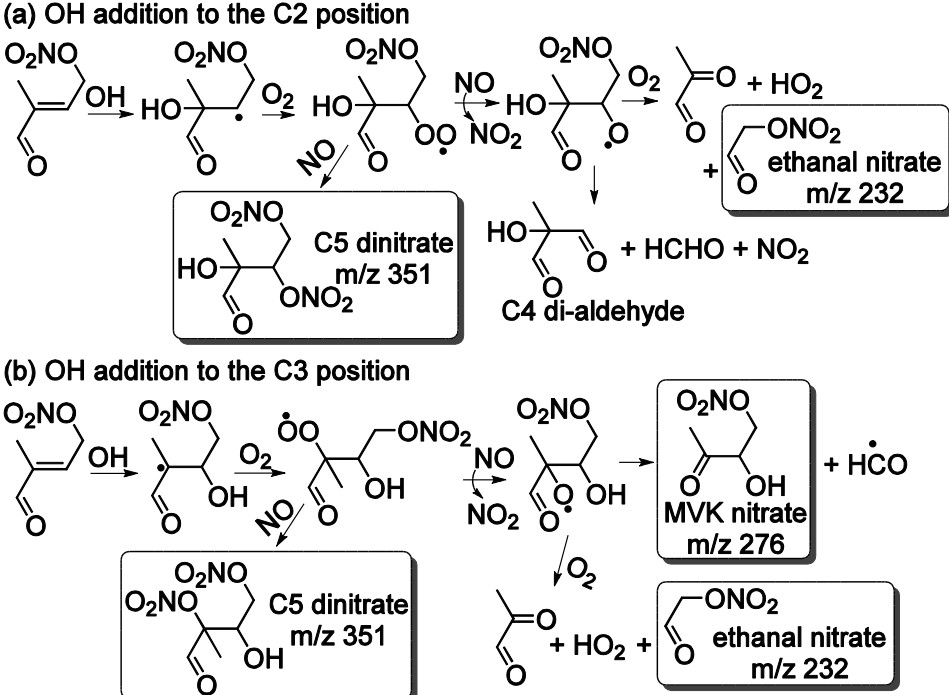


Figure 12. Proposed reaction mechanisms for OH addition to the C2 (a) and C3 (b) position of
the carbonyl nitrate. The compounds in boxes are proposed products as observed by the CIMS
(Fig. 8).






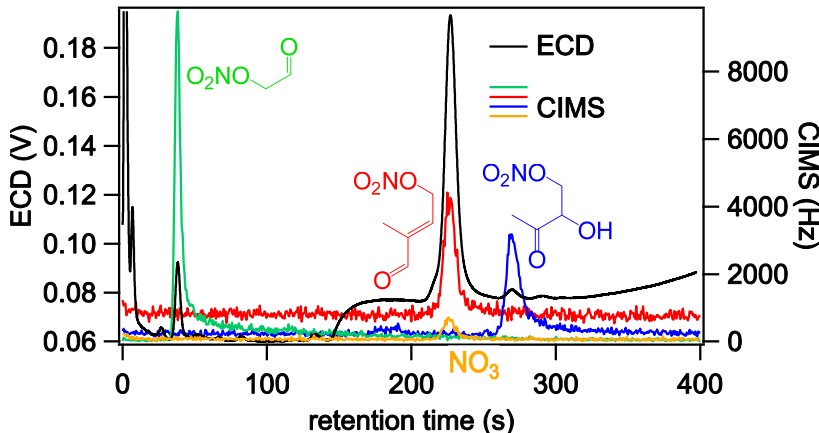


Figure 13. The GC-ECD/CIMS spectra for the carbonyl nitrate (red), MVK nitrate (blue) and
ethanal nitrate (green). The reaction of iodide with the carbonyl nitrate generated $NO_3^-$ ion
(orange). The ECD chromatogram is shown in black.


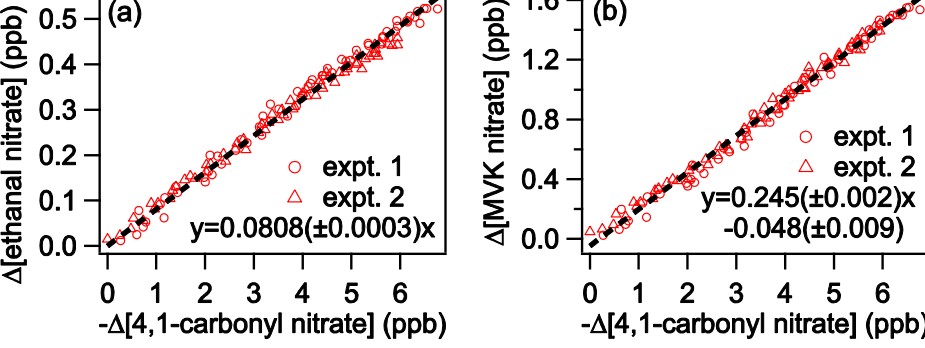


Figure 14. The formation of ethanal nitrate (a) and MVK nitrate (b) relative to the loss of the
isoprene carbonyl nitrate for the OH + carbonyl nitrate oxidation experiments.





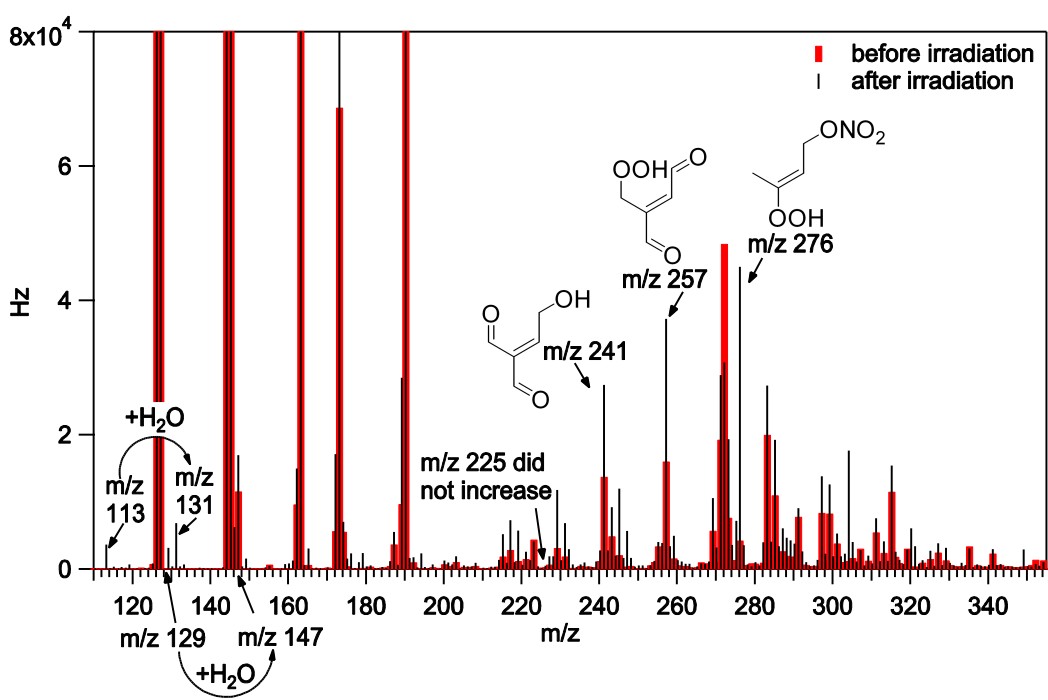


Figure 15. CIMS spectra before (red) and after (black) the photolysis of the isoprene carbonyl
nitrate.




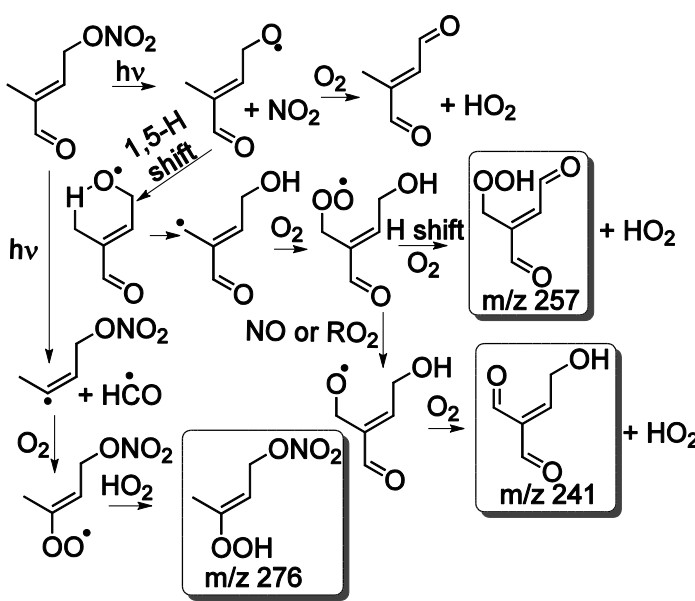


Figure 16. A proposed reaction mechanisms for the carbonyl nitrate photolysis reaction. The
compounds in boxes are proposed products as observed by the CIMS (Fig. 13).