# Peer review of "Photochemical Degradation of Isoprene-derived 4,1-Nitrooxy Enal"

_Atmospheric Chemistry and Physics, 2016_

## Referee Comment (RC1) · J.-F. Müller (Referee) · 18 Feb 2016

This article presents a nice and detailed investigation of the photolysis, ozonolysis and OH-reaction of a conjugated carbonyl nitrate produced in the oxidation of isoprene by $NO_3$ radical. The topic is of great importance to atmospheric chemistry since the formation and fate of organic nitrates play an outstanding role through their influence on the budget of NOx over forested areas. The methods are appropriate and the analysis is sound (with some minor reservations as explained further below). The article is also very well-written, very clear, and appropriately illustrated. Although the focus is on a specific compound which in itself plays probably only a very minor role in the atmosphere, the results regarding the rates of photolysis and reaction with OH and $O_3$ are very likely valid to a broader class of compounds which are important intermediates in the oxidation of isoprene and (no doubt) many other compounds.

[Figure]

**General comment**

Among several interesting findings, this study provides sound evidence that the interaction between the two chromophores in a nitrooxy enal enhances its absorption cross section as well as its photolysis quantum yield, so much so that photolysis is their dominant sink in atmospheric conditions around mid-day. This view was proposed as a general trait for alpha- and beta-nitrooxy carbonyls (Muller et al., 2014), based on the laboratory observation of strongly enhanced photolysis rates (compared to non-conjugated carbonyls) for several keto-nitrates (Suarez-Bertoa et al., 2012) and for several compounds including ethanal nitrate, the simplest aldehyde nitrate (Muller et al., 2014). That this enhancement also exists for nitrooxy enals (or enones) was previously proposed, but lacked experimental proof, which is provided here. The conjugated nature of the compound under consideration is very important given the distinct features of photolysis parameters of enals or enones compared to other carbonyls, and I think this aspect should be acknowledged in the manuscript. Because of very low quantum yields (ca. 0.004), the photolysis of MACR and MVK is almost negligible in spite of their very high cross sections above 300 nm. The presence of the nitrate group is found to increase the quantum yield by two orders of magnitude, to a value of the order of unity (0.28-0.48 in this study). On top of that, the cross sections are also enhanced, as nicely shown in this work. Overall, the presence of the ONO2 group has a much more dramatic impact for the photolysis rates of enals (or enones) than for other carbonyls. For this reason, I recommend that the studied compound should be referred to as an enal in the title and in the abstract.

In addition, the article presents an experimental determination of the OH- and $O_3$-reaction rates of the nitrooxy enal, thereby enabling the estimation of the relative contribution of photolysis and reaction with OH and $O_3$ to the total photochemical sink of this compound. Photolysis is found to be generally dominant during the day. The further degradation mechanism following photolysis or reaction with OH is also explored, and yields of different products are derived. Photolysis is believed to proceed in part

by O–NO2 dissociation, as proposed in Muller et al. (2014), and for some part by C–CHO scission. Interpretation of the CIMS measurements and the derivation of yields is helped by kinetic modelling to account for the losses of the main observed products. The only reservation I have concerns the choice of photolysis rates for those products in this analysis (see further below). But this is only a minor issue which should not affect the main conclusions of the study. I therefore recommend publication in ACP, after the authors take the above considerations into account, and address the following comments.

**Minor comments**

lines 71-78: The interaction between chromophores in nitrooxy carbonyls (i.e. also aldehydes) was found to enhance not only the cross sections but also the quantum yields (Muller et al., 2014). The combined effects on cross sections and quantum yields were observed for ethanal nitrate and for the sum of methyl vinyl ketone nitrate and methacrolein nitrate (MVKNO3 + MACRNO3) of which the measured temporal evolution in the experiment of Paulot et al. (2009) provided constraints on the photolysis parameters. A quantum yield of the order of unity was also proposed for the major nitrooxy enal produced in the oxidation of isoprene by $NO_3$. Its estimated photolysis rate was 5.6 x $10^{-4}$ $s^{-1}$ for a solar zenith angle of 30 degrees, assuming a unity quantum yield and using the cross sections of MACR. As a consequence, photolysis was estimated to outrun OH-oxidation in atmospheric conditions.

line 168: The error bar for the wall loss rate constant appears somewhat optimistic in view of the scatter shown on Fig. 6. How was it derived?

line 175: "... cis isomer was present". I guess you mean "... was formed from the trans isomer", correct?

Figure 7. The caption should tell that the cross sections of the nitrate were obtained in acetonitrile.

line 239: Is the factor 1.7 an average weighted by the irradiance spectrum?

lines 393: Using the photorate of the 4,1-carbonyl nitrate to represent the photolysis loss of ethanal nitrate and the MVK nitrate is not appropriate as those compounds are not conjugated and their absorption cross sections are expected to be much lower in the relevant wavelength range (300-400 nm). For MVKNO3, I recommend to use the cross sections of 3-nitrooxy-2-butanone which are known from Barnes et al. (1993), and a quantum yield of unity since this choice led to best results for MV-KNO3+MACRNO3 evolution in Muller et al. (2014). For ethanal nitrate, the cross sections shown in Fig. 2 in Muller et al. (2014) could be used, as it was also found to give good results against Paulot et al. This update should decrease the calculated photolysis frequencies, especially for MVKNO3. Note that the OH-reaction rate of MV-KNO3 according to Kwok and Atkinson (1995) is 1.3 x $10^{-12}$ $cm^3$ $molec^{-1}$ $s^{-1}$, which might not be entirely negligible.

line 494: As far as photolysis is concerned, I don't really see why unsaturated ketones would be much different from unsaturated aldehydes. The absorption cross sections and quantum yields of MVK and MACR are very similar.

**Technical corrections**

line 106 "derived"

line 213: "were known" –> "are known"

line 214: "we calculated" –> we calculate"

line 220: "introduced" –> "introduce"

line 226: "we calculated that lambda0 was..." –> "we calculate that lambda should be..."

line 273: "multiplying by..."

**References**

Barnes, I., Becker, K. H., and Zhu, T., J. Atmos. Chem., 17, 353-373, 1993.

Kwok, E. S. C., and Atkinson, R., Atmos. Environ., 29, 1685-1695, 1995.

Müller, J.-F., Peeters, J. and Stavrakou, T., Atmos. Chem. Phys., 2497-2508, 2014.

Paulot, F. et al., Atmos. Chem. Phys., 9, 1479-1501, 2009.

Suarez-Bertoa, R. et al., Environ. Sci. Technol., 46, 12502–12509, 2012.
* * *

---

## Referee Comment (RC2) · Anonymous Referee #2 · 17 Mar 2016

The manuscript, "Photochemical degradation of isoprene-derived 4,1-carbonyl nitrate" by Xiong et al. reports on the photolysis rate of the trans-4-1 carbonyl nitrate derived in the atmosphere from the NO3 radical oxidation of isoprene. The manuscript is well written and describes a great deal of well-thought-out work. The main implication of this work is that this conformer of isoprene carbonyl nitrate will have a short lifetime in the atmosphere due mainly to photolysis, with non-negligible contribution from OH oxidation. The work also identifies some of the major byproducts of OH oxidation and photolysis of this compound, thereby improving our understanding of isoprene photochemistry. The work should be published in ACP with a few minor clarifications.

Minor questions/comments/suggestions: I disagree with the "double" and "single" exponential discussion (lines 175-178). That a double exponential will be observed if large amounts of cis is present is unconvincing without additional information and likely

cannot be concluded without knowing the isomerization rate. What is the chamber residence time? A cis-trans equilibrium at some point will be reached. If the rate at which this occurs is instantaneous (or at least much faster than residence time), a single exponential will always be observed because the CIMS only ever sees a mixture of the two isomers. Would you expect a significant difference in the photolysis rate of cis versus trans? If not, does it matter which isomer you are measuring? All that matters for this part of the experiment is the decay rate. If photo lifetime of cis versus trans is very different, you would have to qualify that the 1.3e-5 sec-1 rate is some average of the two isomers.

It would be helpful to know which compounds whose structures are drawn in figures 10 and 15, as well as those boxed in figures 11 and 12 are observed by both GC and CIMS. The CIMS captures signal at nominal masses, therefore, to infer not only molecular composition (CxHyOz) but molecular structure (i.e. identifying functional groups) would impart a certain amount of uncertainty. If only the CIMS without GC is used to infer a compound identity (such as dinitrates which I imagine do not survive GC column), this is worth clarifying. Also, how well can you distinguish MVK nitrate from MACR nitrate with GC /CIMS?

Given CIMS observations of boxed compounds in figures 12 and 16, can you infer branching ratios of OH oxidation paths (a versus b in figure 12) and the two photolysis paths (figure 16).

How were the spectra in figure 3 obtained? Are they of three different samples, one containing pure carbonyl nitrate in solvent, the second pure MACR, and the third pure isopropyl nitrate? If the spectra are of one mixture containing all three compounds, how were the spectra distinguished or attributed to a particular compound? Is each a simulated or calculated spectrum from the observed (the sum of the three spectra shown in figure 3). This needs to be better explained, in particular, for the discussion on lines 142 to 165. This discussion tries to establish that the excitation features of carbonyl nitrate is well understood, that the one near 255 nm is from the nitrate group

and the one near 330 nm is from the aldehyde group. However, there are some aspects of this discussion that are difficult to follow, hence, the argument is not as convincing as can be.

For instance, figure 3 shows isopropyl nitrate along with MACR and carbonyl nitrate, whereas figure 4 shows n-butyl nitrate. Explain why the combination of these 3 compounds was chosen for this part of the study...similarity in structure, overlapping functional groups, etc. Figure 4 and 5 involve calculations...why not include isopropyl nitrate as well? It would make comparison simpler and argument more convincing.

Figure 4 is described as an absorption spectrum. But it looks very different from figure 3. Figure 4 looks more like band strength or absorption lines. Is it possible to simulate actual absorption spectra (one for MACR, one for carbonyl nitrate, one for n-butyl nitrate) given data shown in figure 4 under conditions similar to those in figure 3 and compare that result to figure 3? Would provide stronger support to TDDFT calculation.

Lines 142-144, reads as if authors are saying there is a transition for n-butyl nitrate near 330 nm when there is not. Please re-word. Lines 148-149..."...Earth's surface..." what is the significance of this statement?

Figure 5 and lines 159-165. What is the relevance of including the excitation feature near 210 nm (figure 4) when there is no experimental data (figure 3) to compare against. This spectral region is also "beyond atmospheric relevance" as authors note.

Figure 1. Is the reaction between the NO3 radical and nitroxy peroxy radical the only route to the alkoxy radical, hence carbonyl nitrate? Isn't reaction with RO2 more likely than NO3 to generate the alkoxy given abundance of RO2 in most BVOC rich region? At the very least, RO2 should be included. Rollins et al 2009 ACP (www.atmos-chem-phys.net/9/6685/2009/).

Application to field observation was demonstrated in figure 9. Out of curiosity, is there direct observation of isoprene carbonyl nitrate from the field using CIMS+GC? Spec-

trum or chromatogram or time series or diel average? How abundant is isoprene car-bonyl nitrate considering it is produced at night when loss rate is presumably slow? How well can the CIMS distinguish C5H7NO4 from potential interference due to the isotope of the signal at mz 271. Do you have carbonyl nitrate + NO3 oxidation results, similar to those of OH and photolysis shown here? These would be nice additions to this work, but perhaps saving for separate manuscript.

Figure 4. Many have a difficult time distinguishing red from blue. May help to choose different color scheme. Also, are vertical lines necessary to show this data? The three lines at 210 nm are difficult to distinguish from one another. Perhaps use markers instead? Also, change "1×10ˆexponent" to just "10ˆexponent"

The wall loss rate constant is fairly high compared to the photolysis rate constant. What is the residence time in the 5.2 m long tubing? Is laminar flow maintained? Also, curious if heating the inlet to 50 degC can induce cis-trans isomerization.

Line 235. Why is there no gas phase spectrum? Is it technically challenging? If so, it would be helpful for community to know.
* * *

---

## Author Comment (AC1) · 18 Apr 2016

This article presents a nice and detailed investigation of the photolysis, ozonolysis and OH-reaction of a conjugated carbonyl nitrate produced in the oxidation of isoprene by NO3 radical. The topic is of great importance to atmospheric chemistry since the formation and fate of organic nitrates play an outstanding role through their influence on the budget of NOx over forested areas. The methods are appropriate and the analysis is sound (with some minor reservations as explained further below). The article is also very well-written, very clear, and appropriately illustrated. Although the focus is on a specific compound which in itself plays probably only a very minor role in the atmosphere, the results regarding the rates of photolysis and reaction with OH and O3 are very likely valid to a broader class of compounds which are important intermediates in the oxidation of isoprene and (no doubt) many other compounds.

**General comment**
Among several interesting findings, this study provides sound evidence that the interaction between the two chromophores in a nitrooxy enal enhances its absorption cross section as well as its photolysis quantum yield, so much so that photolysis is their dominant sink in atmospheric conditions around mid-day. This view was proposed as a general trait for alpha- and beta-nitrooxy carbonyls (Muller et al., 2014), based on the laboratory observation of strongly enhanced photolysis rates (compared to nonconjugated carbonyls) for several keto-nitrates (Suarez-Bertoa et al., 2012) and for several compounds including ethanal nitrate, the simplest aldehyde nitrate (Muller et al., 2014). That this enhancement also exists for nitrooxy enals (or enones) was previously proposed, but lacked experimental proof, which is provided here. The conjugated nature of the compound under consideration is very important given the distinct features of photolysis parameters of enals or enones compared to other carbonyls, and I think this aspect should be acknowledged in the manuscript. Because of very low quantum yields (ca. 0.004), the photolysis of MACR and MVK is almost negligible in spite of their very high cross sections above 300 nm. The presence of the nitrate group is found to increase the quantum yield by two orders of magnitude, to a value of the order of unity (0.28-0.48 in this study). On top of that, the cross sections are also enhanced, as nicely shown in this work. Overall, the presence of the ONO2 group has a much more dramatic impact for the photolysis rates of enals (or enones) than for other carbonyls. For this reason, I recommend that the studied compound should be referred to as an enal in the title and in the abstract.

We have changed to refer to this compound as "isoprene nitrooxy enal", or "nitrooxy enal", in the title, abstract and the rest of the manuscript.

In addition, the article presents an experimental determination of the OH- and O3- reaction rates of the nitrooxy enal, thereby enabling the estimation of the relative contribution of photolysis and reaction with OH and O3 to the total photochemical sink of this compound. Photolysis is found to be generally dominant during the day. The further degradation mechanism following photolysis or reaction with OH is also explored, and yields of different products are derived. Photolysis is believed to proceed in part C2 by O–NO2 dissociation, as proposed in Muller et al. (2014), and for some part by C– CHO scission. Interpretation of the CIMS measurements and the derivation of yields is helped by kinetic modelling to account for the losses of the main observed products. The only reservation I have concerns the choice of photolysis rates for those products in this analysis (see further below). But this is only a minor issue which should not affect the main conclusions of the study. I therefore recommend publication in ACP, after the authors take the above considerations into account, and address the following comments.

**Minor comments**
lines 71-78: The interaction between chromophores in nitrooxy carbonyls (i.e. also aldehydes) was found to enhance not only the cross sections but also the quantum yields (Muller et al., 2014). The combined effects on cross sections and quantum yields were observed for ethanal nitrate and for the sum of methyl vinyl ketone nitrate and methacrolein nitrate (MVKNO3 + MACRNO3) of which the measured temporal evolution in the experiment of Paulot et al. (2009) provided constraints on the photolysis parameters. A quantum yield of the order of unity was also proposed for the major nitrooxy enal produced in the oxidation of isoprene by NO3. Its estimated photolysis rate was 5.6 x 10−4 s −1 for a solar zenith angle of 30 degrees, assuming a unity quantum yield and using the cross sections of MACR. As a consequence, photolysis was estimated to outrun OH-oxidation in atmospheric conditions.

On lines 73-80 of the revised manuscript, we have added more clarification on the results reported by the Muller et al. (2014) study, to indicate that the interaction between chromophores can enhance both cross section and quantum yield.

line 168: The error bar for the wall loss rate constant appears somewhat optimistic in view of the scatter shown on Fig. 6. How was it derived?

The reported error is the standard error (s) of the coefficient. For clarification, we now report the result with 95% confidence interval, using $t_{(N-2)}$*s on lines 215-216.

line 175: "... cis isomer was present". I guess you mean "... was formed from the trans isomer", correct?

We have re-worded the reasoning for this part on line 226-238.

Figure 7. The caption should tell that the cross sections of the nitrate were obtained in acetonitrile.

We have included this information in the caption.

line 239: Is the factor 1.7 an average weighted by the irradiance spectrum?

The factor 1.7 is not a weighted average. It is calculated as the average ratio of gas-phase cross section divided by condensed-phase cross section at each wavelength. This is clarified on lines 296-298.

lines 393: Using the photorate of the 4,1-carbonyl nitrate to represent the photolysis loss of ethanal nitrate and the MVK nitrate is not appropriate as those compounds are not conjugated and their absorption cross sections are expected to be much lower in the relevant wavelength range (300-400 nm). For MVKNO3, I recommend to use the cross sections of 3-nitrooxy-2-butanone which are known from Barnes et al. (1993), and a quantum yield of unity since this choice led to best results for MVKNO3+MACRNO3 evolution in Muller et al. (2014). For ethanal nitrate, the cross sections shown in Fig. 2 in Muller et al. (2014) could be used, as it was also found to give good results against Paulot et al. This update should decrease the calculated photolysis frequencies, especially for MVKNO3. Note that the OH-reaction rate of MVKNO3 according to Kwok and Atkinson (1995) is $1.3 \times 10^{-12}$ cm3 molec$^{-1}$ s $^{-1}$ , which might not be entirely negligible.

We calculated the photolysis frequency of 3-nitrooxy-2-butanone using the cross section reported by Barnes et al. (1993) and a unity quantum yield. The result, 4.5E-6 s$^{-1}$ is used as a surrogate for the photolysis frequency of MVKNO3. We calculated the MVKNO3 + OH rate constant as 1.78E-12 cm$^3$molec$^{-1}$s$^{-1}$, based on Kwok and Atkinson (1995). We corrected the MVKNO3 yield using these updated loss rates on lines 468-481.

We calculated that the photolysis frequency for ethanal nitrate is 1.69E-5 s$^{-1}$, using the cross section recommended by Muller el al (2014) and a unity quantum yield. This information is added on lines 464-466.

line 494: As far as photolysis is concerned, I don't really see why unsaturated ketones would be much different from unsaturated aldehydes. The absorption cross sections and quantum yields of MVK and MACR are very similar.

The unsaturated ketones and aldehydes are expected to have similar photochemical properties, given their similar structures, but the ketones may not be as reactive to OH as the aldehydes. We have made the clarification on lines 583-587.

**Technical corrections**
line 106 "derived"
line 213: "were known" –> "are known"
line 214: "we calculated" –> we calculate"

line 220: "introduced" –> "introduce"
line 226: "we calculated that lambda0 was..." –> "we calculate that lambda should be..."
line 273: "multiplying by..."

The above corrections have been made.

**References**

Barnes, I., Becker, K. H., and Zhu, T.: Near UV absorption spectra and photolysis products of difunctional organic nitrates: Possible importance as NO x reservoirs, Journal of Atmospheric Chemistry, 17, 353-373, 10.1007/bf00696854, 1993.
Kwok, E. S. C., and Atkinson, R.: Estimation of hydroxyl radical reaction rate constants for gas-phase organic compounds using a structure-reactivity relationship: An update, Atmospheric Environment, 29, 1685-1695, http://dx.doi.org/10.1016/1352-2310(95)00069-B, 1995.
Müller, J. F., Peeters, J., and Stavrakou, T.: Fast photolysis of carbonyl nitrates from isoprene, Atmospheric Chemistry and Physics, 14, 2497-2508, 10.5194/acp-14-2497-2014, 2014.

The manuscript, "Photochemical degradation of isoprene-derived 4,1-carbonyl nitrate" by Xiong et al. reports on the photolysis rate of the trans-4-1 carbonyl nitrate derived in the atmosphere from the NO3 radical oxidation of isoprene. The manuscript is well written and describes a great deal of well-thought-out work. The main implication of this work is that this conformer of isoprene carbonyl nitrate will have a short lifetime in the atmosphere due mainly to photolysis, with non-negligible contribution from OH oxidation. The work also identifies some of the major byproducts of OH oxidation and photolysis of this compound, thereby improving our understanding of isoprene photochemistry. The work should be published in ACP with a few minor clarifications.

Minor questions/comments/suggestions: I disagree with the "double" and "single" exponential discussion (lines 175-178). That a double exponential will be observed if large amounts of cis is present is unconvincing without additional information and likely cannot be concluded without knowing the isomerization rate. What is the chamber residence time? A cis-trans equilibrium at some point will be reached. If the rate at which this occurs is instantaneous (or at least much faster than residence time), a single exponential will always be observed because the CIMS only ever sees a mixture of the two isomers. Would you expect a significant difference in the photolysis rate of cis versus trans? If not, does it matter which isomer you are measuring? All that matters for this part of the experiment is the decay rate. If photo lifetime of cis versus trans is very different, you would have to qualify that the 1.3e-5 sec-1 rate is some average of the two isomers.

The chamber is operated in a static rather than dynamic mode. The duration of each experiment is about 3 hours. If a cis-trans equilibrium is established instantaneously, a single exponential will be observed. However, we don't expect the cis and trans isomer to differ in photolysis frequency, given they both have the nitrooxy enal structure. Therefore, the measured decay rate should represent the photolysis frequency of the trans precursor in the reaction chamber. We have added this discussion to lines 226-238.

It would be helpful to know which compounds whose structures are drawn in figures 10 and 15, as well as those boxed in figures 11 and 12 are observed by both GC and CIMS. The CIMS captures signal at nominal masses, therefore, to infer not only molecular composition (CxHyOz) but molecular structure (i.e. identifying functional groups) would impart a certain amount of uncertainty. If only the CIMS without GC is used to infer a compound identity (such as dinitrates which I imagine do not survive GC column), this is worth clarifying. Also, how well can you distinguish MVK nitrate from MACR nitrate with GC /CIMS?

The compounds that were observed by both GC and CIMS are now indicated in blue. The structures proposed in figures 10-12 and 15-16 are inferred based on nominal masses. We have added this information to the captions of these graphs.

We are unclear how well the GC/CIMS can distinguish MVK nitrate from MACR nitrate. For this work, we do not expect to have MACR nitrate in the system, because the nitrooxy enal has a secondary carbon at its C3 position, and the OH oxidation reaction cannot add a functional group at this position while still maintaining it as a secondary carbon as in MACR nitrate. Therefore, we infer m/z 276 to be MVK nitrate. We have added this discussion to lines 409-412.

Given CIMS observations of boxed compounds in figures 12 and 16, can you infer branching ratios of OH oxidation paths (a versus b in figure 12) and the two photolysis paths (figure 16).

The branching ratios for the OH oxidation cannot be obtained because the products from H-abstraction pathway were not quantified. For the OH addition pathway, we did quantify two of the products. However, ethanal nitrate is produced from both H abstraction and OH addition pathways (including both (a) and (b) pathways). MVK nitrate is produced in pathway (b) only, but it has ethanal nitrate as byproduct (along with C5 dinitrate), which makes it impossible to determine the branching ratio for pathway (b). We have added this discussion to lines 491-497.

For photolysis pathways, we cannot determine the branching ratio because the photolysis products were identified, but not quantified. This is clarified on lines 549-552.

How were the spectra in figure 3 obtained? Are they of three different samples, one containing pure carbonyl nitrate in solvent, the second pure MACR, and the third pure isopropyl nitrate? If the spectra are of one mixture containing all three compounds, how were the spectra distinguished or attributed to a particular compound?

The spectra were obtained with three different samples, each one containing one pure solute in the acetonitrile solvent. This is clarified on line 99-100.

Is each a simulated or calculated spectrum from the observed (the sum of the three spectra shown in figure 3). This needs to be better explained, in particular, for the discussion on lines 142 to 165. This discussion tries to establish that the excitation features of carbonyl nitrate is well understood, that the one near 255 nm is from the nitrate group C2 and the one near 330 nm is from the aldehyde group. However, there are some aspects of this discussion that are difficult to follow, hence, the argument is not as convincing as can be. For instance, figure 3 shows isopropyl nitrate along with MACR and carbonyl nitrate, whereas figure 4 shows n-butyl nitrate. Explain why the combination of these 3 compounds was chosen for this part of the study...similarity in structure, overlapping functional groups, etc. Figure 4 and 5 involve calculations...why not include isopropyl nitrate as well? It would make comparison simpler and argument more convincing.

The calculation was performed for each spectrum separately. In Fig. 3, we compared isopropyl nitrate and MACR with isoprene nitrooxy enal, because MACR has the enal structure, and isopropyl nitrate has the nitrooxy group, and the combination of these two compounds resembles the nitrooxy enal studied in this work. This explanation was added to lines 102-105 of the revised manuscript.

To better compare the measured UV spectra with the calculated spectra, we have now added calculations for isopropyl nitrate in section 3.2 and 4.1 of the revised manuscript.

Figure 4 is described as an absorption spectrum. But it looks very different from figure 3. Figure 4 looks more like band strength or absorption lines. Is it possible to simulate actual absorption spectra (one for MACR, one for carbonyl nitrate, one for n-butyl nitrate) given data shown in figure 4 under conditions similar to those in figure 3 and compare that result to figure 3? Would provide stronger support to TDDFT calculation.

Fig. 4 shows the theoretical gas phase absorption spectra of the nitrooxy enal, MACR, isopropyl nitrate, and n-butyl nitrate. To accurately capture the broadening of these lines in TDDFT, it is required to consider the effect of the chromophore's vibrational degrees of freedom and/or to include a condensed phase environment that surrounds the chromophore. However, explicit modeling of broadening either due to vibronic interactions or solvent effects is computationally challenging. We believe the analysis of this kind is beyond the scope of the present work. It is also possible to artificially broaden stick spectra with Gaussian or Lorentzian envelopes, to represent collision broadening. However, we do not think that such artificial broadening would provide any additional information, while the presented stick theoretical spectra provide adequate support for our arguments. This discussion is added to lines 165-170.

Lines 142-144, reads as if authors are saying there is a transition for n-butyl nitrate near 330 nm when there is not. Please re-word.

We have re-worded the sentence to refer the 330 nm transition as a transition for the nitrooxy enal only on line 180-182.

Lines 148-149..."...Earth's surface..." what is the significance of this statement? Figure 5 and lines 159-165. What is the relevance of including the excitation feature near 210 nm (figure 4) when there is no experimental data (figure 3) to compare against. This spectral region is also "beyond atmospheric relevance" as authors note.

The theoretical calculations suggest that the nitrooxy group has an electronic transition at 210 nm and 255 nm, but both wavelengths are outside the solar radiation spectrum near the surface. Therefore, we speculate that the isoprene nitrooxy enal absorbs photons primarily through the transition of the enal chromophore, instead of the nirooxy functionality, and the dissociation of the $O-NO_2$ bond likely results from intramolecular energy redistribution. This discussion is added to line 207-212.

Including those transitions gives a clear picture of how the theoretical spectrum of the nitrooxy enal is a function of both nitrooxy absorption and absorption of the enal group, through the comparison with the absorption of the alkyl nitrates and MACR. While though they might not seem directly relevant to the experimental data, they show internal consistency of the simulated spectra. This discussion is further clarified on line 198-200.

Figure 1. Is the reaction between the NO3 radical and nitroxy peroxy radical the only route to the alkoxy radical, hence carbonyl nitrate? Isn't reaction with RO2 more likely than NO3 to generate the alkoxy given abundance of RO2 in most BVOC rich region? At the very least, RO2 should be included. Rollins et al 2009 ACP (www.atmos-chemphys.net/9/6685/2009/).

We have now included $RO_2$ as a second reactant to form nitrooxy alkoxy radicals in Fig. 1.

Application to field observation was demonstrated in figure 9. Out of curiosity, is there direct observation of isoprene carbonyl nitrate from the field using CIMS+GC? Spectrum or chromatogram or time series or diel average? How abundant is isoprene carbonyl nitrate considering it is produced at night when loss rate is presumably slow? How well can the CIMS distinguish C5H7NO4 from potential interference due to the isotope of the signal at mz 271. Do you have carbonyl nitrate + NO3 oxidation results, similar to those of OH and photolysis shown here? These would be nice additions to this work, but perhaps saving for separate manuscript.

To date there is no report on field observations of isoprene carbonyl nitrates using CIMS or GC method. One of the challenges for this type of measurement might be that when iodide-based CIMS is used, the isoprene nitrooxy enal can react with iodide and form $NO_3^-$, instead of nitrate-iodide cluster, and the nitrooxy enal could be detected as $NO_3$ and $N_2O_5$ radicals. In addition, iodide-based CIMS is most sensitive to species with acidic hydrogens, which the enal nitrate does not have. Brown et al. (2009) observed $NO_3$ + isoprene chemistry in Northeast US in the 2004 NEAQS study, and they estimated that the total concentrations of the isoprene-derived nitrate could reach 500 ppt. The carbonyl nitrates are expected to contribute a significant fraction to the total organic nitrates estimated by Brown et al. (2009), but the exact amount cannot be obtained without direct measurement of the carbonyl nitrates. This discussion is added to lines 434-440.

Our CIMS has unit mass resolution. For this work, we used pure isoprene nitrooxy enal as the precursor, which did not introduce much interference at m/z 271. This is clarified on lines 122-124.

Since this work is focused on the photochemistry of the nitrooxy enal, which describes the loss-dominant processes after sunrise, we did not include experiments concerning $NO_3$ oxidation. This is clarified on lines 116-118 of the revised manuscript.

Figure 4. Many have a difficult time distinguishing red from blue. May help to choose different color scheme. Also, are vertical lines necessary to show this data? The three lines at 210 nm are difficult to distinguish from one another. Perhaps use markers instead? Also, change "1×10ˆexponent" to just "10ˆexponent"

Fig. 4 was updated including the changes suggested by the reviewer.

The wall loss rate constant is fairly high compared to the photolysis rate constant. What is the residence time in the 5.2 m long tubing? Is laminar flow maintained? Also, curious if heating the inlet to 50 degC can induce cis-trans isomerization.

The wall loss and photolysis rate constants were obtained with repeated experiments. The radiation inside the chamber is approximately 10% of solar radiation. Therefore, our photolysis rate constant is small, making the wall loss rate constant high compared with photolysis frequency. This information is added to line 218-220.

The residence time in the tubing is around 5 s, and laminar flow is maintained. We have conducted inlet tests for the heated tubing, and we do not expect significant isomerization inside our sampling line. This information is added to line 125-131.

Line 235. Why is there no gas phase spectrum? Is it technically challenging? If so, it would be helpful for community to know.

We were concerned about potentially large wall loss of the organic nitrate inside a small UV cell. Hence, the measurements were performed with nitrate solutions. This is clarified on line 100-102.

**Reference**

[revised manuscript text omitted]